# Spectroscopic evidence of superconductivity pairing at 83 K in single-layer FeSe/SrTiO₃ films

Yu Xu[1,2,6], Hongtao Rong[1,2,6], Qingyan Wang [1,2,6✉], Dingsong Wu[1,2,6], Yong Hu [1,2], Yongqing Cai[1,2], Qiang Gao[1,2], Hongtao Yan[1,2], Cong Li[1,2], Chaohui Yin[1,2], Hao Chen[1,2], Jianwei Huang[1], Zhihai Zhu[1,2], Yuan Huang[1,2], Guodong Liu[1,2,3], Zuyan Xu[4], Lin Zhao[1,2,3✉] & X. J. Zhou [1,2,3,5✉]

Single-layer FeSe films grown on the SrTiO₃ substrate (FeSe/STO) have attracted much attention because of their possible record-high superconducting critical temperature ($T_c$) and distinct electronic structures. However, it has been under debate on how high its $T_c$ can really reach due to the inconsistency of the results from different measurements. Here we report spectroscopic evidence of superconductivity pairing at 83 K in single-layer FeSe/STO films. By preparing high-quality single-layer FeSe/STO films, we observe strong superconductivity-induced Bogoliubov back-bending bands that extend to rather high binding energy ~100 meV by high-resolution angle-resolved photoemission measurements. They provide a new definitive benchmark of superconductivity pairing that is directly observed up to 83 K. Moreover, we find that the pairing state can be further divided into two temperature regions. These results indicate that either $T_c$ as high as 83 K is achievable, or there is a pseudogap formation from superconductivity fluctuation in single-layer FeSe/STO films.

[1] National Lab for Superconductivity, Beijing National Laboratory for Condensed Matter Physics, Institute of Physics, Chinese Academy of Sciences, Beijing, China. [2] University of Chinese Academy of Sciences, Beijing, China. [3] Songshan Lake Materials Laboratory, Dongguan, China. [4] Technical Institute of Physics and Chemistry, Chinese Academy of Sciences, Beijing, China. [5] Beijing Academy of Quantum Information Sciences, Beijing, China. [6] These authors contributed equally: Yu Xu, Hongtao Rong, Qingyan Wang, Dingsong Wu. ✉email: qingyanwang@iphy.ac.cn; lzhao@iphy.ac.cn; XJZhou@iphy.ac.cn

The macroscopic property of a superconductor is characterized by zero resistance and diamagnetism. Electronically, the superconducting state is characterized by the opening of a superconducting gap along the Fermi surface accompanied by the suppression of spectral weight near the Fermi level ($E_F$). In conventional superconductors, there is a good correspondence between the gap opening temperature and the superconducting critical temperature $T_c$ determined directly from transport and magnetic measurements. However, for unconventional superconductors like cuprates, it has been found that the gap opening along the Fermi surface may occur at a significantly higher temperature than $T_c$. Such a gap opening above $T_c$ is attributed to the formation of the pseudogap[1]. The observation of pseudogap is important for understanding high temperature superconductivity in cuprates although its origin remains under debate on whether it is due to superconductivity fluctuation or competing orders[1]. The electron pairing and formation of long-range phase coherence of the Cooper pairs are two essential prerequisites for realizing superconductivity. In conventional superconductors, they occur at the same temperature because of the high superfluid density. But in two-dimensional cuprate superconductors with low superfluid density, the electron pairing temperature may become higher than the phase coherence temperature, giving rise to the formation of pseudogap that corresponds to pre-formed pairing and superconductivity fluctuation above $T_c$[2–4]. On the other hand, symmetry breakings or formation of other competing orders may also result in the formation of an energy gap above $T_c$[1]. In iron-based superconductors, there has been no consensus on whether there is a peudogap formation[5–15]. In single-layer FeSe/STO films, it has been found that, while the critical temperature $T_c$ from the transport and magnetic measurements are scattered[16–21], most of the transport $T_c$ (onset $T_c$ ~40 K)[16–18,21] is obviously lower than that obtained from the spectroscopic $T_c$ (gap opening temperature ~65 K)[22–25]. This raises questions on how high the superconductivity pairing temperature and superconducting critical temperature ($T_c$) can be achieved and whether there is a pseudogap formation in the single-layer FeSe/STO films, as suggested in a related intercalated FeSe superconductor[15] that shares common electronic structures with the single-layer FeSe/STO films[26]. Addressing these issues is important for exploring for high $T_c$ superconductivity and understanding the superconductivity mechanism in iron-based superconductors.

In this work, we report spectroscopic evidence of superconductivity pairing up to 83 K in single-layer FeSe/STO films from high-resolution angle-resolved photoemission (ARPES) measurements. We prepared high-quality single-layer FeSe/STO films that makes it possible to clearly resolve two electron-like Fermi surface around the Brillouin zone corners. In particular, we observed remarkably strong superconductivity-induced Bogoliubov back-bending bands that are standard signature of superconductivity pairing. The direct analysis of the Bogoliubov back-bending bands, combined with analyses on the energy gap and spectral weight, indicates that the superconductivity pairing can persist up to 83 K in the single-layer FeSe/STO films. The superconductivity pairing state can be further divided into two temperature regions of 64–83 K and below 64 K according to their distinct behaviors. We propose the 64–83 K region may correspond to superconductivity fluctuation while the temperature region below 64 K corresponds to the realization of superconducting phase coherence in single-layer FeSe/STO films.

## Results

### Observation of superconductivity-induced Bogoliubov back-bending bands

We prepared high-quality single-layer FeSe/STO films[27,28] and carried out high-resolution ARPES measurements on detailed temperature evolution of their electronic structures. Figure 1 shows Fermi surface mapping around M3($-\pi$, $-\pi$) point (Fig. 1a) and representative band structures measured along two momentum cuts at a low temperature and a high temperature. The constant energy contour around M3 ($-\pi$, $-\pi$) at a binding energy of 18 meV (Fig. 1b) shows that there are two ellipse-like electron pockets perpendicular to each other although their spectral intensity varies along the pockets because of the photoemission matrix element effects. In addition to the central electron pockets, a large white circle can also be observed in Fig. 1b. In the band structures measured at 83 K (Fig. 1c) and 20 K (Fig. 1d) along the momentum Cut 1, two electron-like bands can be seen from the raw data and from the second derivative data (see Supplementary Fig. 1), consistent with the previous report[29]. While these bands cross the Fermi level ($E_F$) at 83 K (Fig. 1c), strong suppression of the spectral weight at the Fermi level can be seen from the image at 20 K which indicates a gap opening (Fig. 1d). Moreover, two new sets of bands appear at 20 K on both sides of the bands. The band splitting and the appearance of new bands can be seen more clearly in Fig. 1e which is obtained from dividing the image at 20 K (Fig. 1d) by the one at 83 K (Fig. 1c) to highlight the net temperature-induced change. These observations indicate that at 20 K the sample enters a superconducting state where superconducting gaps open for both of the electron-like bands, accompanied by a loss of spectral weight at the Fermi level. The new bands observed in Fig. 1d are from the formation of the Bogoliubov back-bending bands. Figure 1f, g shows the extracted band structures from Fig. 1c, d, respectively. Taking the normal state band structures in Fig. 1f and getting the superconducting gap size for the two bands in Fig. 1d (9 meV for the inner band and 13 meV for the outer band), we can extract the band structure in the superconducting state, $E_k$, by the BCS formula $E_k = -[(\xi_k)^2 + (\triangle_k)^2]^{1/2}$ where $\xi_k$ represents the normal band structure and $\triangle_k$ is the superconducting gap size. The band structures thus extracted agree well with the measured ones in Fig. 1g (see Supplementary Fig. 2). In the same manner, taking the superconducting state band structures in Fig. 1g, we can also extract the band structure in the normal state by the same BCS formula. The extracted normal state band structures thus agree well with the measured ones in Fig. 1f (see Supplementary Fig. 2). These results further support that the sample at 20 K is in the superconducting state. It is noted that the Bogoliubov back-bending bands are surprisingly robust and can be observed at a high binding energy up to 80 meV (Fig. 1d, e), and even the band splitting can also be resolved in the Bogoliubov back-bending bands, as shown in Fig. 1h. Such extended Bogoliubov back-bending bands at 20 K can be used to obtain the unoccupied band structure up to 80 meV above the Fermi level in the normal state according to the above BCS formula (see Supplementary Fig. 2) which is very similar to the calculated band structure above $E_F$[30]. The extra large white circle seen in Fig. 1b can now be understood as due to the formation of the extended Bogoliubov back-bending bands. The observation of a full white circle also indicates that the formation of the Bogoliubov back-bending bands occurs along the entire electronic pockets, further reinforcing the superconductivity pairing nature of the observed gap opening.

The observation of clear band splitting makes it possible to establish quantitatively an overall electronic structure of the single-layer FeSe/STO films from the band structure measurements along a number of momentum cuts that form the Fermi surface mapping (Fig. 1a). The Fermi surface consists of two ellipse-like electron pockets which are centered around M3 point and perpendicular to each other (black and blue ellipses in Fig. 1a), consistent with the presence of two Fe in one unit cell.

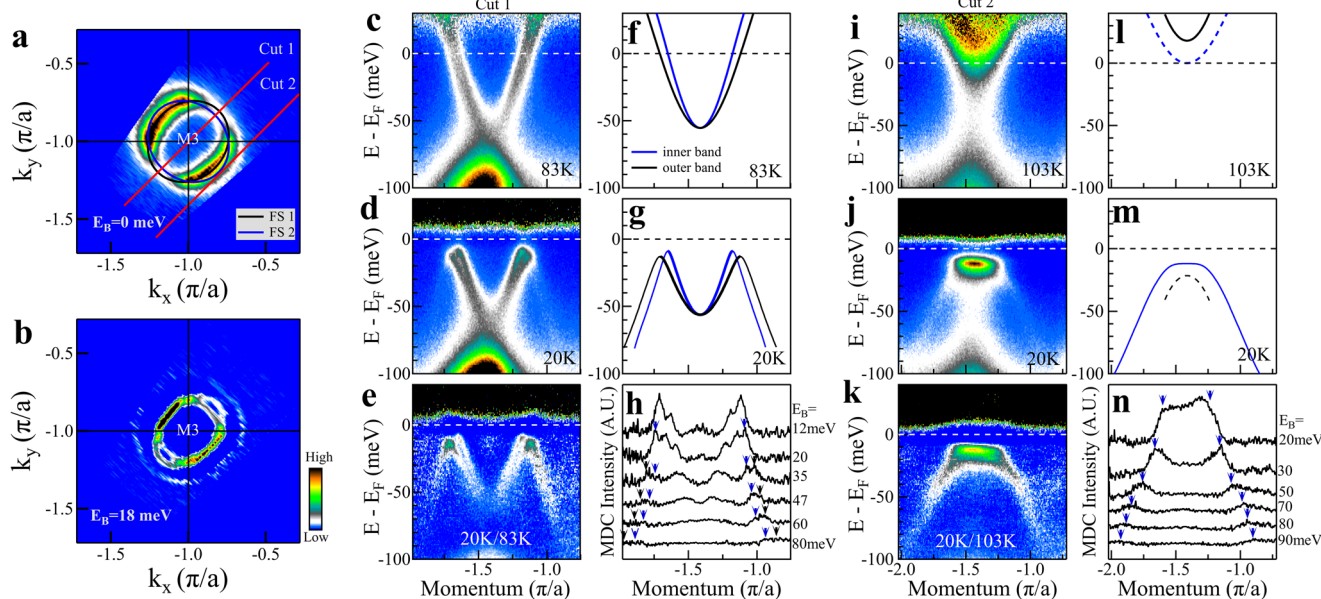

**Fig. 1 Observation of clear band splitting and strong superconductivity-induced Bogoliubov back-bending bands in single-layer FeSe/STO films.**
**a** Fermi surface mapping near M3 (−π, −π) measured at 20 K. It is obtained by integrating the spectral weight within [−5, 5] meV energy window with respect to the Fermi level. Two electron-like Fermi surface sheets are resolved marked by black line (FS 1) and blue line (FS 2). **b** Second derivative constant energy contour near M3 (−π, −π) at a binding energy of 18 meV. Band structure measured along the momentum Cut 1 at 20 K (**d**) and 83 K (**c**), respectively. Each image is divided by the corresponding Fermi distribution function. The location of the momentum Cut 1 is shown in (**a**). **e** Photoemission image obtained from dividing the data at 20 K (**d**) by the one at 83 K (**c**). **f** Two electron-like bands obtained from (**c**). The inner and outer bands come from the FS2 and FS1 Fermi surface sheets in (**a**), respectively. **g** Two electron-like bands and the superconductivity-induced Bogoliubov back-bending bands obtained from (**d**). **h** Representative MDCs of the photoemission image in (**e**) at different binding energies. The splitting of the Bogoliubov back-bending bands is marked by two colored arrows. Band structure along the momentum Cut 2 measured at 20 K (**j**) and 103 K (**i**), respectively. Each image is divided by the corresponding Fermi distribution function. The location of the momentum Cut 2 is shown in (**a**). **k** Photoemission image obtained from dividing the data at 20 K (**j**) by the one at 103 K (**i**). **l** Two electron-like bands corresponding to (**i**). The band marked by solid black line is obtained from (**i**). The band marked by dashed blue line is expected from the overall measured Fermi surface and band structure. **m** Two bands corresponding to (**j**). The band marked by solid blue line is obtained from (**j**). The band marked by dashed black line is extracted from the band at 103 K (solid black line in (**l**)) using the BCS formula. **n** Representative MDCs of the photoemission image in (**k**) at different binding energies. The blue arrow marks the peak position of Bogoliubov back-bending band.

We chose a unique momentum Cut 2 (Fig. 1a) to measure the band structures at a high temperature (103 K, Fig. 1i) and a low temperature (20 K, Fig. 1j). For this momentum Cut 2, in the normal state, one electron band is observed above the Fermi level (black line in Fig. 1l) while the other band is expected from the band structure established in this work that is above the Fermi level with its bottom barely touching the Fermi level (dashed blue line in Fig. 1l). This band is not observed in the measured data in Fig. 1i because it is weak presumably due to photoemission matrix element effect. Remarkably, in the superconducting state at 20 K (Fig. 1j), a strong flat band and two sets of Bogoliubov back-bending bands appear below E_F while those bands above the Fermi level are fully suppressed because of the Fermi cutoff at 20 K. Again, these bands become more pronounced in Fig. 1k which is obtained from dividing the superconducting data in Fig. 1j by the normal state data in Fig. 1i. Similar to the results from Cut 1, the band structure at 103 K in Fig. 1l and the observed bands at 20 K in Fig. 1m can be well connected by the BCS formula calculations (taking a gap size of 12 meV, see Supplementary Fig. 3). The observed flat band and the Bogoliubov back-bending bands (blue solid line in Fig. 1m) are from the lower band in the normal state (dashed blue line in Fig. 1l). The other band expected in the superconducting state (black dashed line in Fig. 1m) from the normal state band (black solid line in Fig. 1l) is not resolved in Fig. 1j. The selection of the momentum Cut 2 is advantageous in that in the normal state all the bands are above the Fermi level with only one dominant band

observed whereas at low temperature all the bands below $E_F$ are from superconductivity-induced Bogoliubov back-bending bands (see Supplementary Fig. 4). In the present case, in the superconducting state, the signal below $E_F$ is dominated by only one band. To the best of our knowledge, such strong Bogoliubov back-bending bands are observed for the first time in the single-layer FeSe/STO films mainly due to much improved sample quality. While superconductivity is related with a superconducting gap opening along the Fermi surface, the gap opening alone is not necessarily related to superconductivity because it may also originate from other competing orders like charge density wave[31]. The Bogoliubov back-bending band, on the other hand, is a unique signature of superconductivity pairing. It therefore provides a good opportunity to study electron pairing and the nature of the energy gap in single-layer FeSe/STO films.

**Temperature dependence of the energy gap and the associated spectral weight.** Since superconductivity is intimately related to the gap opening along the Fermi surface, we first examine on the temperature dependence of the energy gap in single-layer FeSe/STO films. Figure 2a shows the photoemission spectra (energy distribution curves, EDCs) measured at different temperatures at the Fermi momentum of the outer Fermi surface, $k_{F\_OR}$, on the momentum Cut 1. Sharp coherence peaks develop with decreasing temperature. To extract the energy gap, we follow the usual procedure to get symmetrized EDCs (Fig. 2b)[32], and obtain

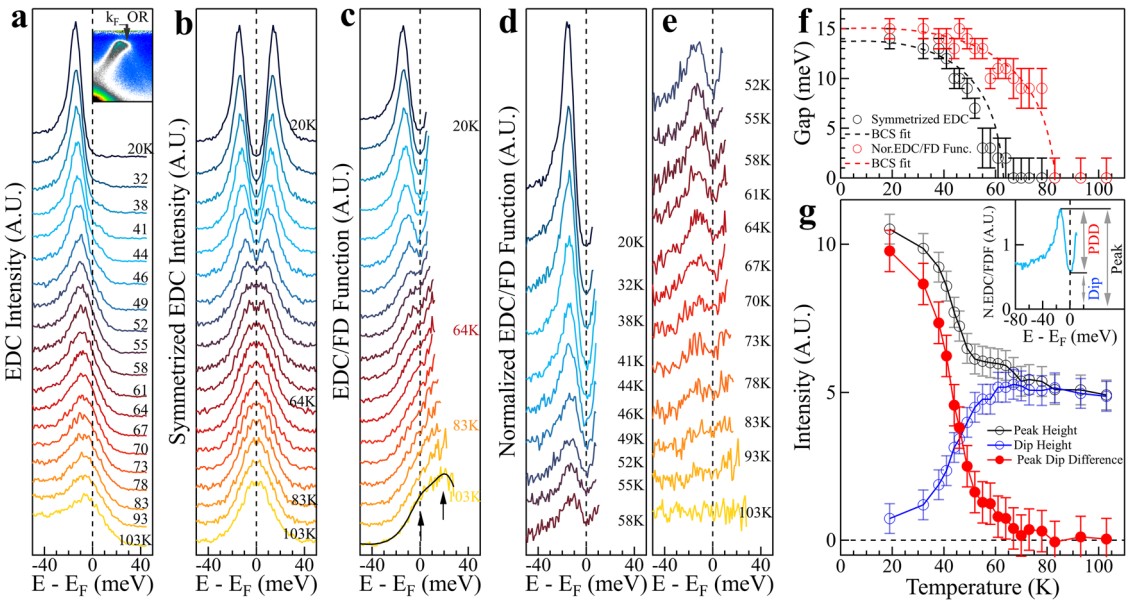

**Fig. 2 Temperature dependence of the energy gap and the associated spectral weight along a Fermi surface. a** Photoemission spectra (EDCs) measured at different temperatures at the Fermi momentum on the right side of the outer Fermi surface ($k_F$_OR) along the momentum Cut 1. The location of the momentum Cut 1 is illustrated in Fig. 1a and the location of the Fermi momentum $k_F$_OR is shown in the upper-right inset. **b** The corresponding symmetrized EDCs at different temperatures obtained from (**a**). **c** EDCs at different temperatures obtained from dividing the original EDCs in (**a**) by their corresponding Fermi distribution functions (FD Function). The solid black line at the bottom represents the fitted curve of the 103 K data by two Lorentzians which correspond to two bands, the outer-right band is at the Fermi level while the other inner-right one lies ~20 meV above the Fermi level as indicated by two arrows. **d, e** EDCs at different temperatures obtained from dividing the EDCs in (**c**) by the fitted curve at 103 K. For clarity, the obtained normalized EDCs are displayed in (**d, e**). **f** Temperature dependence of the energy gap. The black circles show the energy gap obtained by picking the peak position of the symmetrized EDCs in (**b**). The red circles show the gap obtained by picking the peak position of the normalized EDCs in (**d, e**). The dashed lines show the fitted curve of the energy gap by the BCS formula. **g** Temperature dependence of the spectral weight obtained from the normalized EDCs in (**d, e**). As shown in the upper-right inset, here we take three spectral intensities: the height of the peak below the Fermi level, the height of the dip at the Fermi level and the difference between the peak height and the dip height (PDD). The corresponding three spectral intensities as a function of temperature are shown by the empty black circles, empty blue circles and the red solid circles, respectively. Error bars reflect the uncertainty in determining the EDC peak position and spectral intensity.

the gap size ($\Delta$) from the peak position as plotted in Fig. 2f (black circles). From the symmetrized EDC (Fig. 2b), the dip structure at the Fermi level represents a gap opening; it disappears around 64 K for the Fermi momentum of the outer Fermi surface, $k_F$_OR, that appears to be consistent with the previous results[23–25]. Similar procedure is also used to the Fermi momentum of the inner Fermi surface, $k_F$_IL; in this case, the dip structure at the Fermi level in the symmetrized EDCs persists to a much higher temperature near 93 K (see Supplementary Fig. 5b). The inconsistence of the gap closing temperature obtained from the two Fermi momenta is apparently not reasonable. Now with two bands clearly resolved, we come to realize that such a standard procedure of extracting the gap size from the EDC symmetrization is not reliable at high temperature for the single-layer FeSe/STO films. In the case of the inner Fermi surface, $k_F$_IL, two peaks in EDCs below $E_F$ can be clearly observed at low temperature because of another band from the outer Fermi surface (see Supplementary Fig. 5a). At high temperature, the dip at the Fermi level is not due to the gap opening but is an artifact caused by another EDC peak at a higher binding energy (see Supplementary Fig. 5b). This is understandable because the EDC symmetrization procedure is valid under an assumption of a particle-hole symmetry at the Fermi momentum and it works only for a single band[32]. In our case, the two bands are quite close around the Fermi level that they will influence each other, thus invalidating the EDC symmetrization procedure in the energy gap determination at high temperature.

The EDC symmetrization procedure is used to extract gap size because it provides a convenient way to remove the Fermi distribution function to get the spectral function[32]. When it becomes unreliable for the single-layer FeSe/STO films because of two co-existing bands, a straightforward way to get the spectral function is to directly divide each EDC in Fig. 2a by their corresponding Fermi distribution function[3,4]. The EDCs thus obtained are shown in Fig. 2c. The other band from the inner Fermi surface can be detected at high temperature (103 K) which is ~20 meV above the Fermi level (Fig. 2c). Comparing the EDCs in Fig. 2c and in Fig. 2b, it becomes clear that the symmetrized EDCs do not represent intrinsic spectral function at high temperatures. Therefore, the EDC symmetrization approach is not a reliable way to extract the energy gap at high temperature for the single-layer FeSe/STO films, even for the outer Fermi surface where only one EDC peak is observed below $E_F$ (Fig. 2a). We note that such an argument also applies to the case that the two bands are not well-resolved due to the sample quality or instrumental resolution[23–25].

In order to get the gap information from the intrinsic spectral functions in Fig. 2c, we further divided the EDCs at low temperatures by the EDC at 103 K and the obtained normalized EDCs are shown in Fig. 2d, e. To highlight the temperature-induced changes and remove the effect of background, it is a standard procedure to normalize the low temperature spectra by a spectrum at a high temperature in the normal state in spectroscopic measurements[33,34]. A sharp EDC peak and a clear dip at $E_F$ can be seen from the EDCs at low temperatures; the peak-dip structure gets weaker with increasing temperature but is visible up to ~83 K (Fig. 2d, e). From these normalized EDCs, the temperature dependence of the energy gap and spectral weight

can be obtained. The presence of a dip at $E_F$ signals a gap opening; the gap size is determined by the position of the EDC peak below the Fermi level. The gap opening thus obtained (red circles in Fig. 2f) extends to a higher temperature (~83 K). To illustrate the spectral weight change with temperature, we define three quantities: the peak height, the dip height, and the peak-dip height difference, as shown in the inset of Fig. 2g. The dip height is a good measure of the spectral weight change with temperature at the Fermi level due to the opening of an energy gap. The peak-dip height difference can be used as a measure of superconductivity[34]. The extracted results are shown in Fig. 2g. It is clear that, with decreasing temperature, the EDC peak gets sharper; the spectral weight at $E_F$ gets suppressed, and the peak-dip difference starts to show up at ~83 K and shoots up dramatically at low temperature. Furthermore, to check on the feasibility of our approach, we also carried out similar analysis on the data from $Bi_2Sr_2CaCu_2O_8$ (Bi2212) which has one main band along the measured momentum cut[35]. The energy gap obtained from the EDC symmetrization procedure and from the normalized EDCs is consistent (see Supplementary Fig. 6).

**Temperature dependence of the Bogoliubov back-bending bands and the associated spectral weight.** Now we switch our analysis to the data taken at different temperatures along the momentum Cut 2. As shown in Fig. 1, since the normal state bands at high temperature are above $E_F$ (Fig. 1i, l), all the bands

observed below the Fermi level at low temperature are superconductivity-induced Bogoliubov back-bending bands. The central flat band shows a strong spectral intensity while the two side bands can be observed clearly which extend to a rather high binding energy (~100 meV) (Fig. 1j) which is consistent with the theoretical predictions of the superconducting pairing features in single-layer FeSe/SrTiO₃ films up to energies close to 100 meV[30]. These bands are also free from the complication of two co-existing bands encountered in the measurements along the momentum Cut 1 (Fig. 2). All these make the momentum Cut 2 measurements ideal to study the gap opening and super-conductivity pairing in single-layer FeSe/STO films. Figure 3a shows photoemission images measured along the momentum Cut 2 at different temperatures divided by the image at 103 K to highlight the Bogoliubov back-bending bands and their changes with temperature. We find that the Bogoliubov back-bending bands on both sides get weaker with increasing temperature, but this signal is visible at high temperature 78–83 K (Fig. 3a). This observation provides a direct evidence of superconductivity pairing at 83 K in single-layer FeSe/STO films.

To analyse the central flat band, Fig. 3b, c shows the normalized EDCs taken at the central momentum (marked by the arrow in 20 K panel of Fig. 3a) at different temperatures. The original EDCs and the procedures to get the EDCs shown in Fig. 3b, c are presented in Supplementary Fig. 7. A strong and sharp EDC peak develops below $E_F$ at 20 K with a spectral dip at

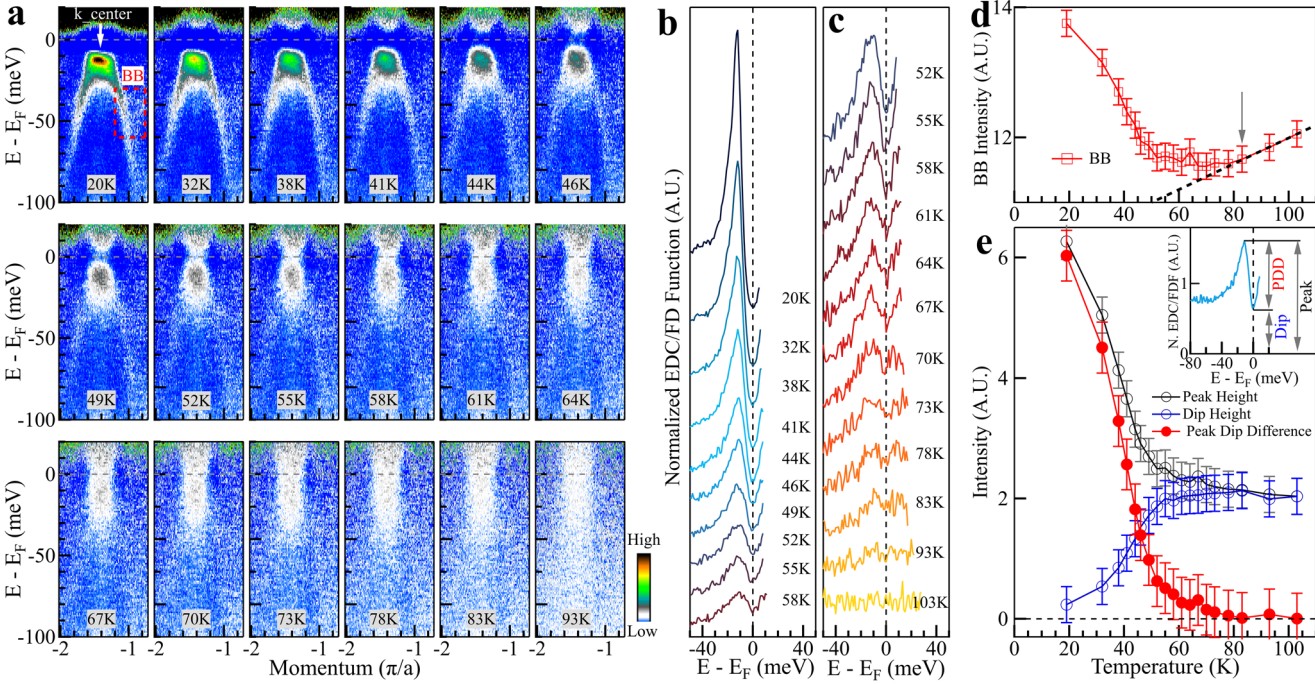

**Fig. 3 Temperature dependence of the band structure, the photoemission spectra, and the associated spectral weight measured along the momentum Cut 2. a** Photoemission images measured along the momentum Cut 2 at different temperatures. The location of the momentum Cut 2 is shown in Fig. 1a. These images are obtained by first dividing the original images measured at different temperatures with their respective Fermi distribution functions, and then dividing the images by the data at 103 K. Superconductivity-induced Bogoliubov back-bending is strong that extends to rather high binding energy up to 100 meV at low temperatures. The signature of the Bogoliubov back-bending band is clear at 67–70 K and is still visible at high temperature 78–83 K. **b, c** Normalized EDCs at the momentum k_center. The location of the momentum is marked in the top-left panel in (**a**). The normalized EDCs are obtained by the same procedure as in Fig. 2d, e. The original EDCs and the EDCs divided by their respective Fermi distribution functions are shown in Supplementary Fig. 7. **d** Temperature dependence of the spectral weight of the superconductivity-induced Bogoliubov back-bending band. The area where the spectral weight is obtained is marked in the upper-left 20 K panel in (**a**) by the dashed red box. The procedure to extract the spectral weight is described in Supplementary Fig. 8. **e** Temperature dependence of the spectral weight obtained from the normalized EDCs in (**b, c**). As shown in the upper-right inset, we take three spectral intensities: the height of the peak below the Fermi level, the height of the dip at the Fermi level and the difference between the peak height and the dip height (PDD). The corresponding three spectral intensities as a function of temperature are shown by the empty black circles, empty blue circles and the red solid circles, respectively. Error bars reflect the uncertainty in determining the spectral intensity.

the Fermi level (Fig. 3b). Such a peak-dip structure gets weaker with increasing temperature but can still be visible up to 83 K (Fig. 3b, c). The resultant peak height, dip height, and the peak-dip difference are shown in Fig. 3e and it is clear that the peak-dip structure starts to develop at 83 K.

To quantitatively analyze the Bogoliubov back-bending bands on the two sides, we integrated the spectral weight over the energy-momentum window shown by the BB box in the 20 K panel of Fig. 3a that covers a portion of the Bogoliubov bands (see Supplementary Fig. 8). Figure 3d shows the obtained spectral weight change with temperature. Since the spectral weight of the newly-developed Bogoliubov bands sits on a tail of a strong MDC peak (see Supplementary Fig. 8b), the integrated spectral weight shown in Fig. 3d consists of two contributions: the Bogoliubov bands and the central MDC peak. The spectral weight increase with increasing temperature in the temperature range of 83–103 K is due to slight MDC broadening while the spectral weight gain at low temperature region is due to the appearance of the Bogoliubov back-bending bands. These two competing contributions give a measured spectral weight variation with temperature in Fig. 3d. A reflection point at 83 K is consistent with the direct observation of the Bogoliubov back-bending bands in Fig. 3a.

**Summary of various temperature-induced changes.** Figure 4 summarizes the above results of the photoemission spectra, the

energy gap, the suppression of the spectral weight at $E_F$ and the Bogoliubov back-bending bands measured along two momentum cuts (Figs. 2 and 3). The spectral weight suppression at $E_F$ (Fig. 4c) and the peak-dip height difference (Fig. 4d) show a high-degree agreement between the two momentum cuts measurements. These combined results provide a consistent picture that the temperature dependence of the electronic properties can be divided into three regions based on their different behaviors. In the high temperature region 83–103 K (region 3 in Fig. 4), there is no peak-dip structure observed in EDCs (Figs. 2d, e, 3b, c, and 4a), no gap opening (Figs. 2f and 4b), no indication of Bogoliubov backing-bending bands (Fig. 3a), so the sample is in normal state. Below 83 K, the peak-dip structure in EDCs appears (Figs. 2d, e, 3b, c, and 4a), the gap opens (Figs. 2f and 4b) accompanied by the spectral weight suppression at the Fermi level (Figs. 2g, 3e, and 4c), and most importantly, the direct observation of the Bogoliubov back-bending bands (Fig. 3a), all indicating that the sample is in superconductivity pairing state. Upon a careful inspection, the temperature region below 83 K appears to consist of two sub-regions: 64–83 K (region 2 in Fig. 4) and below 64 K (region 1 in Fig. 4). In the region 2, 64–83 K, even though all the superconductivity pairing signatures are present, these features are weak and they do not show strong temperature dependence. In a strong contrast, below 64 K in region 1, all the features become clear and particularly they all exhibit a dramatic temperature dependence.

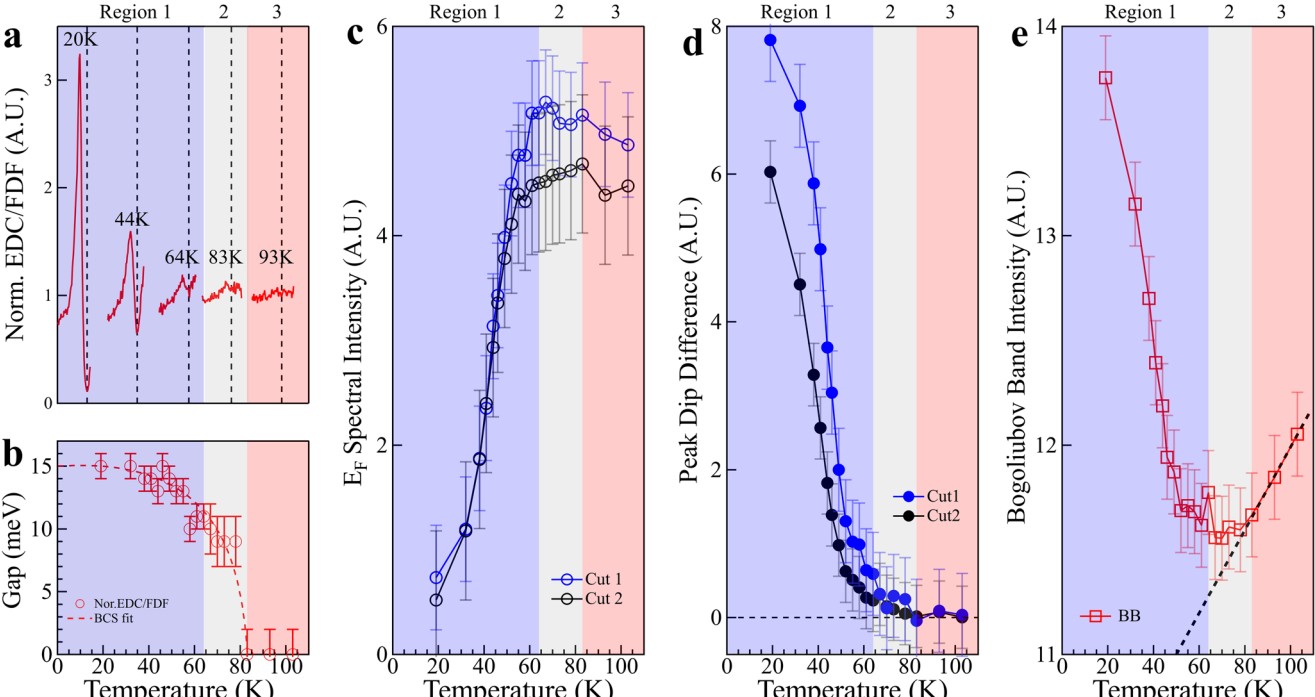

**Fig. 4 Summary of various temperature-induced changes in single-layer FeSe/STO films that can be categorized into three temperature regions.** All the results can be divided into three regions: in region 3, 83–103 K, the sample is in normal state with no Bogoliubov back-bending band, zero gap and no suppression of spectral weight at the Fermi level. In region 2, 64–83 K, there are signatures of Bogoliubov back-bending band, gap opening, and suppression of spectral weight at the Fermi level. But the changes with temperature in this temperature region is small. In region 1, below 64 K, Bogoliubov back-bending band, gap opening, and suppression of spectral weight at the Fermi level are obvious and change dramatically with the temperature decrease. **a** Representative normalized EDCs in three regions. These EDCs are taken from Fig. 3b, c. The vertical dashed lines show the Fermi level for each EDC. **b** Temperature dependence of the energy gap along a Fermi surface. The data are from Fig. 2f. **c** Temperature-induced spectral weight change at the Fermi level for the momentum Cut 1 (blue empty circles) and Cut 2 (black empty circles). The data are from Fig. 2g for Cut 1 and from Fig. 3e for Cut 2; they are normalized for comparison. **d** Temperature dependence of the difference between the peak height and the dip height from the normalized EDCs for Cut 1 (blue solid circles) and Cut 2 (black solid circles). The data are from Fig. 2g for Cut 1 and from Fig. 3e for Cut 2; they are normalized for comparison. **e** Temperature dependence of the spectral weight of the superconductivity-induced Bogoliubov back-bending band. The data are from Fig. 3d. Error bars reflect the uncertainty in determining the EDC peak position and spectral intensity.

## Discussion

The observation of 83 K superconductivity pairing temperature in single-layer FeSe/STO films has signification implications. On the one hand, it may point to a high critical temperature $T_c$. In ARPES measurements on FeSe-related superconductors, $A_xFe_{2-y}Se_2$ (A = K, Rb, Cs, Tl and etc.) superconductors with a $T_c$ ~32 K[36–38] and (Li, Fe)OHFeSe superconductor with a $T_c$ of 42 K[26], the gap opening temperatures are consistent with the transport and magnetic $T_c$, indicating no pseudogap formation in these superconductors. If this is also the case for single-layer FeSe/STO films, one may expect a $T_c$ as high as ~83 K can be achieved. On the other hand, if 83 K represents the electron pairing temperature but superconductivity occurs at a lower temperature, this would point to a formation of pseudogap. In this case, our results would provide the first spectroscopic evidence on the pseudogap formation in single-layer FeSe/STO films. Such a pseudogap originates from superconductivity fluctuation because electron pairing occurs at 83 K. To pin down on whether there is a pseudogap formation in the single-layer FeSe/STO films, $T_c$ determination from transport or magnetic measurements is necessary. The transport measurement on single-layer FeSe/STO films is challenging because the samples are fragile that can easily be damaged when exposed to air. When transport measurements are performed ex situ on samples that are capped with protection layers, the measured $T_c$(onset) is around ~40 K[16–18]. However, it is unclear whether the superconducting properties are altered or not during this protection and measurement process because they depend sensitively on the electron doping level and sample quality[23]. On the other hand, in situ transport measurements are preferred, but the obtained $T_c$s are rather scattered, varying from 109 K[19] to ~40 K[21]. The existing transport results are not sufficient to help us determine the nature of the observed energy gap and further efforts are needed to overcome the transport measurement problems to obtain an intrinsic $T_c$ in the single-layer FeSe/STO films. It is desirable to carry out spectroscopic and transport measurements on the same single-layer FeSe/STO film with high quality and appropriate electron doping level in order to resolve the $T_c$ controversy and pin down on the existence and nature of the pseudogap.

The FeSe layer is a basic building block of the FeSe-based superconductors. It has been found that $T_c$ of the FeSe-related superconductors is sensitive not only to the doping of the FeSe layers but also to the interaction between the FeSe layers. The pristine bulk FeSe exhibits a $T_c$ of ~8 K[39] which can be enhanced to 37–40 K under high pressure[40,41]. Doping bulk FeSe with electrons through gating or surface deposition leads to a $T_c$ increase up to 48 K[42–44]. Intercalation of bulk FeSe gives a $T_c$ up to 50 K at ambient pressure[45–48] and 55 K at high pressure[49]. In the FeSe-related superconductors, $T_c$ of 30–46 K is achieved in $A_xFe_{2-y}Se_2$ (A = K, Rb, Cs, Tl and etc.)[50–52] and $T_c$ of 42 K is realized in (Li, Fe)OHFeSe at ambient pressure[53] and 50 K at high pressure[54]. These results indicate that transport $T_c$ as high as 55 K is already achievable in FeSe-related superconductors[52]. It is also found that, in the intercalated FeSe superconductors, $T_c$ increases with the increasing distance between the FeSe layers[45–48], suggesting that the "intrinsic $T_c$" of an isolated FeSe layer where the interlayer interaction is fully removed may be even higher than 55 K. Magnetic measurement provides a $T_c$ value of 65 K in the single-layer FeSe/STO film[20] which is close to that determined from spectroscopic measurements[23–25]. This indicates that it is possible to achieve a macroscopic $T_c$ at 65 K, and the region 1 below 64 K may represent a true superconducting state with the full realization of phase coherence of the Cooper pairs. In this case, because of its different behaviors, the 64–83 K region 2 may

be attributed to the superconductivity fluctuation and the single-layer FeSe/STO films become similar to the cuprate superconductors with a strong superconductivity fluctuation[3,4]. We find that the electronic behaviors of the single-layer FeSe/STO films are similar to that observed in Bi2212 (see Supplementary Fig. 6) where the gap opening temperature extends to ~140 K although its $T_c$ is 91 K. In the temperature range of 91–140 K, there is a pseudogap formation that can be related to superconductivity fluctuation[4]. Below $T_c$ = 91 K, superconductivity is realized in Bi2212 due to the formation of long-range phase coherence of the Cooper pairing. In this case, the peak height, dip height and peak-dip difference all exhibit dramatic change with temperature (see Supplementary Fig. 6h) that are similar to those observed in single-layer FeSe/STO films (Figs. 2g and 3e).

In summary, we prepared high-quality single-layer FeSe/STO films that make it possible to resolve clearly the band splitting and in particular the superconductivity-induced Bogoliubov back-bending bands from the high-resolution ARPES measurements. The Bogoliubov back-bending bands provide a decisive signature of superconductivity pairing. We have identified an indication of superconductivity pairing at 83 K directly from the Bogoliubov back-bending bands in single-layer FeSe/STO films. We find that the superconductivity pairing region can be further divided into two regions: 64–83 K region and the region below 64 K. There are two possibilities to assign the two regions. In the first case, the former 64–83 K region may correspond to superconductivity fluctuation while the latter region below 64 K may correspond to the phase coherence of Cooper pairs. In the second case, actual long-range pairing is realized in both regions. Further work is needed to distinguish between these two scenarios. The identification of superconductivity pairing at 83 K in single-layer FeSe/STO films is significant because it either indicates that $T_c$ as high as 83 K is achievable in iron-based superconductors, or it provides spectroscopic evidence on the existence of pseudogap in single-layer FeSe/STO films, making it similar to high temperature cuprate superconductors.

## Methods

**Preparation of single-layer FeSe/STO films**. High-quality single-layer FeSe/STO films were grown by molecular beam epitaxy (MBE) method on single-crystal 0.5 wt % Nb-doped $SrTiO_3$(001) substrates (Shinkosha STEP substrates), similar to a procedure reported before[16,27,28]. The base pressure of the MBE chamber is $1 \times 10^{-10}$ mbar. $TiO_2$-terminated atomically flat $SrTiO_3$(001) surface was prepared by degassing at ~600 °C for several hours and subsequently annealed at ~1030 °C for 1 h. Ultrahigh-purity selenium (99.9999%) and iron(99.995%) were coevaporated onto the $SrTiO_3$ substrate from two Knudsen cells with a flux ratio of ~10:1. The temperature of $SrTiO_3$ substrate was held at ~500 °C during growth as measured by the pyrometer, and the growth rate was ~0.1 ML/min. To transfer the samples from the MBE chamber to our ARPES measurement chamber, the prepared films were covered by an amorphous Se capping layer. The Se layer was removed by heating up the samples to ~530 °C for 2 h before ARPES measurements.

**High-resolution ARPES measurements**. High-resolution ARPES measurements were carried out on our laboratory system equipped with a Scienta Omicron DA30 electron energy analyzer[55]. We used a helium discharge lamp as a light source that can provide a photon energy of h$\nu$ = 21.218 eV (He I). The energy resolution was set at 4 meV and the angular resolution was ~0.3°. The Fermi level was referenced by measuring on a clean polycrystalline gold that was electrically connected to the sample. The sample was measured in vacuum with a base pressure better than $5 \times 10^{-11}$ mbar.

## Data availability

All data are processed by using Igor Pro 8.02 software. All data needed to evaluate the conclusions in the paper are available within the article and its Supplementary Information files. All raw data generated during the current study are available from the corresponding author upon reasonable request.

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

## Acknowledgements

We thank financial support from the National Key Research and Development Program of China (Grant No. 2016YFA0300300, 2016YFA0300600, 2017YFA0302900, 2018YFA0704200, 2018YFA0305600 and 2019YFA0308000), the National Natural Science Foundation of China (Grant No. 11888101, 11922414 and 11874405), the Strategic Priority Research Program (B) of the Chinese Academy of Sciences (Grant No. XDB25000000 and XDB33010000), the Youth Innovation Promotion Association of CAS (Grant No. 2017013 and 2019007), and the Research Program of Beijing Academy of Quantum Information Sciences (Grant No. Y18G06).

## Author contributions

Y.X., H.T.R, Q.Y.W., and D.S.W.contribute equally to this work. X.J.Z., L.Z., Q.Y.W., and Y.X. proposed and designed the research. Y.X., Q.Y.W., Y.H., H.T.R contributed to MBE thin film preparation. Y.X., Y.H., D.S.W., Y.Q.C, H.C, G.D.L, Z.Y.X. and X.J.Z. contributed to the development and maintenance of Laser-ARPES and MBE-STM systems. Y.X., H.T.R, and D.S.W. carried out the ARPES experiments. Y. X., H.T.R, D.S.W, Q.Y.W., Y.Q.C, Q.G., H.T.Y., J.W.H, C.L., C.H.Y., H.C., Z.H.Z., Y.H., L.Z., and X.J.Z. analyzed the data. X.J.Z., L.Z., and Y.X. wrote the paper. All authors participated in discussions and comments on the paper.

## Competing interests

The authors declare no competing interests.
