## [Peer Review File · Nature Communications]

REVIEWER COMMENTS

Reviewer #1 (Remarks to the Author):

In this paper, single-layer FeSe films grown on the SrTiO₃ substrate (FeSe/STO) are investigated by ultra-high resolution and high-precision photoelectron spectroscopy. A Bogoliubov back-bending band was found that extends to nearly $E=100$ meV. The superconducting pair is also found to exist up to 83 K. The temperature dependence of the superconducting gap is also investigated. From the temperature dependence of the superconducting gap, the superconducting transition temperature is found to be 63 K and the superconducting pair is found to exist up to 83 K.

First of all, experiments in the low temperature phase are beautiful, but, in the high temperature phase, there are not enough experimental facts to support the fierce claim.

The low temperature data is very beautiful. It is also consistent with previous reports (Ref. 29 in the paper) that two electron bands are visible.

The backbending band itself was claimed in the past by previous reports (Ref. 29 in the paper). The "we observe for the first time..." in the abstract is an overstatement

The title is inappropriate. The spectroscopic evidence is that there is a gap (up to about 67K), and the existence of a superconducting pair (precursor) up to 83K is only an interpretation.

It is not so surprising that the T_c of FeSe/STO is scattered because each researcher has his or her own way of making materials. In this study, author should write the reason why the sample is not non-uniform. I am wondering T_c and $T(\text{superconducting pair})$ is always 65K and 83K, when authors check the reproducibility.

About Bogoliubov back-bending band

It is better to assign this band to some other band than to assign it to the Bogoliubov back-bending band for the reasons described below. I think that it should not be assigned as a Bogoliubov back-bending band.

1. Since the energy region in which the Bogoliubov band is observed is $|E-\mu| < |\Delta|$, it is unlikely that the Bogoliubov band will be observed in a deep energy region that greatly exceeds $|\Delta| \sim 10$ meV.

2. The energy dependence of the Bogoliubov back-bending band is different from the data we know. Normally, it weakens sharply away from the Fermi level, but there is almost no intensity change in this paper. The authors have to explain this. For example, Matsui, PRL, 21702, 90 (2003) has an equation for the intensity dependence, which should be used to explain it.

3. In the raw data of MDC, back bending (shoulder structure of the main peak of Fig. S7b) can be seen only up to about 50K.

4. It is hard to say that back bending exists up to 83K even with the band dispersion shown in Fig. 3a.

5. In Fig. 3d, the background on the high temperature side is extrapolated with a dotted line, but it is

not valid because the error bar is large and there are only two data points above 83K. Therefore, the development of the backbending below the arrow (83K) is doubtful.

6. Why can't I see two bands in cut2? Authors said it is not observed because of the selection rule. If so, what kind of the selection rule?

7. The Bogoliubov back-bending band should be visible even above Fermi level. But in this paper we could not observe this back-bending band.

There is a problem with data analysis regarding the superconducting gap.

1. The data show that the pseudogap is open above T_c . However, 84 K is an overstatement.

2. I understood well that symmetrization of the spectrum is not a good idea in this paper. However, I don't understand why authors need to use the symmetrized spectrum to obtain $T_c=64K$. In fact, the temperature at which the gap is closed differs between the inner band and the outer band, if we use symmetrized spectra. (Fig.S4)

3. I understood well that the most accurate way is to divide the superconducting gap by FD, rather than the symmetrized spectra. However, it(Fig. 2b) closes the superconducting gap at around 58 K. This is the temperature that is obvious to everyone. This is the temperature at which everyone can clearly say that the superconducting gap is closed around 58K.

4. It is an arbitrary data arrangement to divide the spectrum divided by FD by the spectrum of 103K. This is not a good idea because there is another band that rises to the right nearby (Fig.2c), and if you divide by this, it naturally looks like there is a pseudo-gap, as in Fig.2d, e. I think this is an artificial gap.

Reviewer #2 (Remarks to the Author):

In the manuscript entitled "Spectroscopic Evidence of Superconductivity Pairing at 83 K in Single-Layer FeSe/SrTiO₃ Films" by Xu et al., the authors present high-resolution ARPES studies of single layer FeSe/STO that provide the first ARPES evidence of a superconductivity-related gap at temperatures as high as 83 K in this system. Moreover, they observe the clear formation of Bogoliubov quasiparticle bands up to surprisingly high binding energies of near 100 meV. Lastly, they report the temperature dependence of the observed gap which they find to deviate greatly from the standard BCS behavior in the region between 64-83 K. The authors suggest that this region could either correspond to a pseudogap due to superconducting fluctuating Cooper pairs or to a true ordered state, therefore providing ARPES proof of superconductivity with $T_c=83K$ in FeSe/STO.

The mechanism and the associated phenomenology of high- T_c superconductivity in single layer FeSe/STO is arguably at the forefront of current research in the field. Despite the intensive research, reports of T_c 's surpassing the 60-65 K limit are sparse and indeed ARPES observations confirming superconductivity at even higher temperatures have been highly anticipated since the reported superconductivity up to 100 K in this system a few years back that was based on resistivity data.

The work by Xu et al. is therefore timely, groundbreaking and has potential to attract broad interest. Their ARPES results capturing so clearly the formation of Bogoliubov bands over such a wide range of energies and temperatures are very convincing. The evidence of gap formation at temperatures above 64 K seems solid. I am not against proposing that a pseudogap may be the reason for the anomalous temperature dependence reported for "region 2" but I am not convinced either that this is more probable than actual long-range pairing in this region. In this respect, the authors could consider improving the discussion section. Regardless of the cuprate analogy, the reported results are interesting on their own right and I believe they will inspire further theoretical and experimental studies for this system.

There are a few points that need to be addressed. These concern mostly clarifying aspects of the procedure followed by the authors regarding the non-symmetrized ARPES spectra, providing more details for the BCS fits and making contact with available theory predictions that are directly relevant to the presented data.

- It appears that the gap magnitude (for the same k-point) is systematically found to be larger when using the "normalized EDC/FD" method instead of the symmetrized EDC method, even for low temperatures (e.g. see Fig.2f).

How do the authors justify this difference? For example, it has been shown that the estimated gap from ARPES should not depend on whether the symmetrization procedure of the ARPES spectra is followed or not, since the overall shape of the spectra may be asymmetric but the coherence peaks remain particle-hole symmetric (see [PRB 98, 094509 (2018)]).

Is the same effect observed for points on Cut 2? It would be helpful if the authors included a plot similar to that of Fig.2f for a point on the Cut 2 path.

- In Figs 2g,3e and 4c,d the uncertainty in the the points located at the 64-83 K region (region 2) seems significant. The "Peak Dip Difference" in Fig.2g for example may very well be zero above 64 K. Similarly, the "Peak Height" and "Dip Height" may be converging to a single value at around 5 for high temperatures (as the blue and black lines seem to do in Fig.2g) but these two curves could also be bifurcating to two different values at high temperature, given the quite large allowed error for the data. While it does seem that there exists a gap feature, the $T > 64$ K data of Fig.2e look "too noisy" and quite difficult to reliably extract from them the above mentioned quantities such as the "Peak Height" etc.

Perhaps additional plots including more data from Fermi surface points and/or more details on the post processing of the data could help further strengthen the authors case.

- In the supplementary, the authors include their results for Bi2212 in order to provide a validity check to their "normalized EDC/FD" method. Their Bi2212 data are indeed in good agreement with previous findings.

However and in contrast to Fig.2f, Fig.S5g shows that the gap converges to the same value at low temperatures. In fact, both methods shown in Fig.S5g give very similar data even for temperatures up to T_c , again in contrast to what is observed in FeSe/STO. Regardless of these differences, the three quantities shown in Fig.S5h seem to follow a T-dependence trend that is very similar to that observed for FeSe/STO (Figs.2g, 3e).

How do the authors explain the similarity between Figs.2g,3e and Fig.S5h but the difference between Fig.2f and Fig.S5g?

- The authors say: "For this momentum Cut 2, in the normal state, one electron band is observed above the Fermi level (black line in Fig. 1l) while the other band is expected from the established electronic structure that is above the Fermi level with its bottom barely touching the Fermi level (dashed blue line in Fig. 1l)."

The electronic energy band structure of single layer FeSe/STO has proven particularly difficult to calculate with standard bandstructure methods. Do the authors have any particular reference in mind? Especially since according to their ARPES data, these bands are not found by experiment. What is the established electronic structure for single-layer FeSe/STO that the authors mention?

- Connected with the above, the authors mention that they calculated the superconducting band structure using the BCS formula $E_k = \sqrt{\xi_k^2 + \Delta_k^2}$. I understand that they set Δ_k as an isotropic gap with chosen value so as to fit the experiment. What is the form of ξ_k ? Is this calculated and how? How do the authors' fits and experimental data compare with previously calculated ARPES data (see e.g. [New J. Phys. 18 022001 (2016)], [PRB 97, 060501 (2018)]) ?

- Superconducting pairing features in FeSe/STO up to energies close to 100meV were predicted theoretically in [PRB 97, 060501 (2018)]. I suggest that the authors compare their findings with previous such predictions regarding ARPES spectra of FeSe/STO.

- It needs to be pointed out that the pseudogap in the case of the cuprates has been studied exhaustively and its existence established by many different probes. A characteristic feature of the pseudogap in the ARPES results of cuprates is the observed Fermi arc where only part of the Fermi surface is gapped out above T_c . Another well known feature is the almost temperature independent gap above T_c seen by tunneling spectroscopy.

Are the authors aware of or expect such similar signatures also in the case of FeSe/STO?

- Is there any reasoning as to why region 2 cannot correspond to actual long range pairing, given that previous experiments ([Nature Mater. 14, 285-289 (2015)]) provide support for T_c 's up to 100 K?

Response to Reviewer's Comments

Reviewer #1 (Remarks to the Author):

In this paper, single-layer FeSe films grown on the SrTiO₃ substrate (FeSe/STO) are investigated by ultra-high resolution and high-precision photoelectron spectroscopy. A Bogoliubov back-bending band was found that extends to nearly $E=100$ meV. The superconducting pair is also found to exist up to 83 K. The temperature dependence of the superconducting gap is also investigated. From the temperature dependence of the superconducting gap, the superconducting transition temperature is found to be 63 K and the superconducting pair is found to exist up to 83 K.

First of all, experiments in the low temperature phase are beautiful, but, in the high temperature phase, there are not enough experimental facts to support the fierce claim.

The low temperature data is very beautiful. It is also consistent with previous reports (Ref. 29 in the paper) that two electron bands are visible.

The backbending band itself was claimed in the past by previous reports (Ref. 29 in the paper). The "we observe for the first time..." in the abstract is an overstatement

The title is inappropriate. The spectroscopic evidence is that there is a gap (up to about 67K), and the existence of a superconducting pair (precursor) up to 83K is only an interpretation.

It is not so surprising that the T_c of FeSe/STO is scattered because each researcher has his or her own way of making materials. In this study, author should write the reason why the sample is not non-uniform. I am wondering T_c and $T(\text{superconducting pair})$ is always 65K and 83K, when authors check the reproducibility.

About Bogoliubov back-bending band

It is better to assign this band to some other band than to assign it to the Bogoliubov back-bending band for the reasons described below. I think that it should not be assigned as a Bogoliubov back-bending band.

1. Since the energy region in which the Bogoliubov band is observed is $|E-\mu| < |\Delta|$, it is unlikely that the Bogoliubov band will be observed in a deep energy region that greatly exceeds $|\Delta| \sim 10$ meV.

2. The energy dependence of the Bogoliubov back-bending band is different from the data we know. Normally, it weakens sharply away from the Fermi level, but there is almost no intensity change in this paper. The authors have to explain this. For example, Matsui, PRL, 21702, 90 (2003) has an equation for the intensity dependence, which should be used to explain it.

3. In the raw data of MDC, back bending (shoulder structure of the main peak of Fig. S7b) can be seen only up to about 50K.

4. It is hard to say that back bending exists up to 83K even with the band dispersion shown in Fig. 3a.

5. In Fig. 3d, the background on the high temperature side is extrapolated with a dotted line, but it is not valid because the error bar is large and there are only two data points above 83K. Therefore, the development of the backbending below the arrow (83K) is doubtful.

6. Why can't I see two bands in cut2? Authors said it is not observed because of the selection rule. If so, what kind of the selection rule?

7. The Bogoliubov back-bending band should be visible even above Fermi level. But in this paper we could not observe this back-bending band.

There is a problem with data analysis regarding the superconducting gap.

1. The data show that the pseudogap is open above T_c . However, 84 K is an overstatement.

2. I understood well that symmetrization of the spectrum is not a good idea in this paper. However, I don't understand why authors need to use the symmetrized spectrum to obtain $T_c=64K$. In fact, the temperature at which the gap is closed differs between the inner band and the outer band, if we use symmetrized spectra. (Fig.S4)

3. I understood well that the most accurate way is to divide the superconducting gap by FD, rather than the symmetrized spectra. However, it(Fig. 2b) closes the superconducting gap at around 58 K. This is the temperature that is obvious to everyone. This is the temperature at which everyone can clearly say that the superconducting gap is closed around 58K.

4. It is an arbitrary data arrangement to divide the spectrum divided by FD by the spectrum of 103K. This is not a good idea because there is another band that rises to the right nearby (Fig.2c), and if you divide by this, it naturally looks like there is a pseudo-gap, as in Fig.2d, e. I think this is an artificial gap.

Response to Reviewer #1

We thank Reviewer #1 for the careful reviewing of our paper and his/her constructive comments and suggestions to improve our paper.

1. The back-bending band itself was claimed in the past by previous reports (Ref. 29 in the paper). The "we observe for the first time..." in the abstract is an overstatement

We are well aware that the back-bending band is observed in the past by previous reports. In addition to Ref. 29 as the Referee mentioned, it was reported in our own papers that are the very first two ARPES papers on single-layer FeSe/SrTiO₃ films (Refs. 22 and 23). In the abstract, we stated that "we observe for the first time strong superconductivity-induced Bogoliubov back-bending bands *that extend to rather high binding energy ~100 meV* by high-resolution angle-resolved photoemission measurements." There is no overstatement because no one has reported before such an observation of back-bending bands that *extend to rather high binding energy ~100 meV*.

2. The title is inappropriate. The spectroscopic evidence is that there is a gap (up to about 67K), and the existence of a superconducting pair (precursor) up to 83K is only an interpretation.

The superconductivity pairing is associated with two phenomena: gap opening and the formation of back-bending band. The latter is more unique than the former because gap opening may be caused by orderings other than superconductivity. Because we observe very strong Bogoliubov back-bending band, it provides us a new definitive benchmark of superconductivity pairing, and the back-bending band is observed up to 83 K (Fig. 3a). The gap size, when analyzed in a right way as is done in the present paper (Fig. 2f), is also consistent with the superconductivity pairing up to 83 K. We do not think that our title is inappropriate because all our results (Fig. 3 and Fig. 4) provide a consistent conclusion that superconductivity pairing occurs up to 83 K in our single-layer FeSe/STO films.

3. It is not so surprising that the T_c of FeSe/STO is scattered because each researcher has his or her own way of making materials. In this study, author should write the reason why the sample is not non-uniform. I am wondering T_c and T (superconducting pair) is always 65K and 83K, when authors check the reproducibility.

The T_c of FeSe/STO can be different because it relies sensitively on the electron doping level that depends on the sample preparation condition and post-annealing condition, as we pointed out clearly in our earlier paper (Ref. 23). When the preparation and post-annealing conditions are fixed, as we did in the present work (the detailed description of the preparation and post-annealing of single-layer FeSe/STO films is given in the Supplementary Materials), the samples and the measured results are highly reproducible.

About Bogoliubov back-bending band

It is better to assign this band to some other band than to assign it to the Bogoliubov back-bending band for the reasons described below. I think that it should not be assigned as a Bogoliubov back-bending band.

1. Since the energy region in which the Bogoliubov band is observed is $|E-\mu| < |\Delta|$, it is unlikely that the Bogoliubov band will be observed in a deep energy region that greatly exceeds $|\Delta| \sim 10$ meV.

The bands we observed can be assigned to superconductivity-induced Bogoliubov back-bending band because they are consistent with all the behaviors of the Bogoliubov back-bending bands.

It is not true that “Since the energy region in which the Bogoliubov band is observed is $|E-\mu| < |\Delta|$, it is unlikely that the Bogoliubov band will be observed in a deep energy region that greatly exceeds $|\Delta| \sim 10$ meV”, as the Referee stated. As shown in Fig. R1, from the simulated superconducting single-particle spectral function, it is clear that, although the superconducting gap size is $\Delta=10$ meV, the Bogoliubov back-bending band can extend to much high energy, like 80 meV. This can also be seen from direct experimental measurement. In high temperature superconductor $\text{Bi}_2\text{Sr}_2\text{CaCu}_2\text{O}_8$, when the superconducting gap is ~ 10 meV, the Bogoliubov back-bending band can be observed up to 50 meV [see Fig. 2 in W. T. Zhang et al., Phys. Rev. B 85, 064514 (2012), Ref. 33 in revised version].

Fig. R1. Simulated superconducting single-particle spectral function. The superconducting gap is $\Delta=10$ meV. It is clear that the Bogoliubov back-bending band can extend to much high energy up to 80 meV.

2. The energy dependence of the Bogoliubov back-bending band is different from the data we know. Normally, it weakens sharply away from the Fermi level, but there is almost no intensity change in this paper. The authors have to explain this. For example, Matsui, PRL, 21702, 90 (2003) has an equation for the intensity dependence, which should be used to explain it.

It is not true that “*but there is almost no intensity change in this paper*”, as stated by the Referee. Fig. R2a shows the band structure measured along the momentum Cut 2 at 20 K. The image is divided by the corresponding Fermi distribution function. The band below E_F marked by blue solid line is from superconductivity-induced Bogoliubov back-bending band. There is a dramatic spectral intensity change with energy, as shown by EDCs (energy distribution curves) and MDCs (momentum distribution curves) for the selected 5 typical points (A, B, C, D and E in Fig. R2a) on the back-bending band. From the photoemission spectra (EDCs) in Fig. R2c, it is clear that the spectral intensity drops rapidly with energy from A, B to C points. It looks like that there is little intensity change with binding energy for the Bogoliubov back bending bands from C to D to E. This is due to the fact that the back-bending bands sit on a shoulder of a big background that comes from the central band, as shown by the MDCs at three different binding energies in Fig. R2d. The net spectral weight of the back-bending band still decreases with increasing binding energy, as shown in Fig. R2e. Overall, from A to E point, there is a dramatic spectral weight decrease with increasing binding energy that is consistent with the behavior expected with the usual Bogoliubov back-bending band.

Furthermore, the back-bending bands we observed are consistent with the behaviors expected from the BCS theory. As suggested by the Referee, we have carried out similar analysis as done in the *Matsui, PRL, 21702, 90 (2003)* paper. According to the BCS theory, the coherence factors can be calculated from the energy bands in the normal-state and in the superconducting-state in the following way: $|u_k|^2 = 1 - |v_k|^2 = \frac{1}{2} \left(1 + \frac{\xi_k}{E_k} \right)$ where ξ_k and E_k are the energy of the normal state band and Bogoliubov back-bending band, respectively. Fig. R2b shows the calculated $|u_k|^2$ and $|v_k|^2$ obtained this way.

We also make a quantitative analysis on the intensity change of the back-bending band. We plot EDCs in Fig.R2c and MDCs in Fig. R2d,e to extract the spectral weight of the back-bending band at 5 typical points on the back-bending band. After proper renormalization of the spectral intensity, we find that the intensity change of the back-bending band as a function of the binding energy follows closely with the obtained coherence factor $|u_k|^2$.

Therefore, it is not true that “*Since the energy region in which the Bogoliubov band is observed is $|E - \mu| < |\Delta|$, it is unlikely that the Bogoliubov band will be observed in a deep energy region that greatly exceeds $|\Delta| \sim 10 \text{ meV}$* ”, as the Referee stated. It is not true either that “*but there is almost no intensity change in this paper*”, as stated by the Referee. The back-bending bands we observed are consistent with all the behaviors of the Bogoliubov back-bending bands, expected from the BCS theory. There is no doubt that the back-bending bands we observed are from superconductivity-induced Bogoliubov back-bending bands.

To clarify this point, we have added Fig. R2 and the above discussions in the Supplementary Materials.

Fig. R2. Analysis of the Bogoliubov back-bending band. (a) Bogoliubov back bending band formed at 20 K for the momentum Cut 2. The black dashed line represents the initial band in the normal state while the blue solid line represents the back-bending band. (b) Coherence factor deduced from the normal state band and the back-bending band at 20 K in (a). The intensity of the back-bending band from EDC analysis (c) and MDC analysis (e) is plotted. The intensity for point A at the Fermi momentum is taken as 0.5 to normalize the EDC intensity for the point B and C. The intensity of point C is taken as the same from both the EDC and MDC analysis to normalize the spectral intensity of the D and E points. (c) EDCs along cuts #1, #2 and #3 as shown in (a). The spectral weight of the back-bending band is marked. (d) Momentum distribution curves (MDCs) along cuts #4, #5 and #6 as shown in (a). (e) Same MDCs as in (d) but are plotted with an offset for clarity. The spectral weight of the back-bending band is marked.

3. In the raw data of MDC, back bending (shoulder structure of the main peak of Fig. S7b) can be seen only up to about 50K.
4. It is hard to say that back bending exists up to 83K even with the band dispersion shown in Fig. 3a.
5. In Fig. 3d, the background on the high temperature side is extrapolated with a dotted line, but it is not valid because the error bar is large and there are only two data points above 83K. Therefore, the development of the backbending below the arrow (83K) is doubtful.

It is well-known that the Bogoliubov back-bending band is already a subtle and weak feature induced by superconductivity. It gets even weaker with increasing temperature. It is the first time that we can observe such clear back-bending bands that can extend up to 100 meV in single-layer FeSe/SrTiO₃ films. This is because we have greatly improved the sample quality and have taken high quality ARPES data. The observation of the back-bending band can be seen directly from Fig. 3a that persists well above 50 K. In Fig. S8b in revised version, we tried to make a quantitative analysis on the spectral weight of the back-bending band by integrating the spectral weight over the energy-momentum window shown by the BB box. The obtained result in Fig. 3d indicates that there is a change at ~ 83 K that is consistent with Fig. 3a and other analyses. We think that the

Referee should look at Fig. 3a for the temperature dependence of the back-bending band. Coming to Fig. S8b in revised version, we believe that the result from quantitative analysis by spectral weight integration is more objective and reliable and it is not true that “back bending (shoulder structure of the main peak of Fig. S7b) can be seen only up to about 50K” as the Referee stated.

In addition to the direct inspection of the back-bending band in Fig. 3a, and its quantitative analysis shown in Fig. S8b and Fig. 3d, strong evidence of superconductivity pairing up to 83 K can be seen directly from temperature-dependent EDCs. As shown in Fig. 2e and Fig. 3c, peak-dip structure starts to develop at 83K and gets more and more obvious with decreasing temperature. This systematic temperature evolution provides clear evidence of superconductivity pairing at 83 K in spite of their weak feature.

6. Why can't I see two bands in cut2? Authors said it is not observed because of the selection rule. If so, what kind of the selection rule?

In ARPES measurements, the measured intensity $I(\mathbf{k}, \omega) \sim A(\mathbf{k}, \omega) |M_{f,i}^k|^2 f(\omega)$ depends on the matrix element $|M_{f,i}^k|^2$ that relies on the electron momentum, and the energy and polarization of the incoming photons. It is common in ARPES experiments that such a matrix element effect may result in suppression of certain bands related with their orbital nature. We observed two electron pockets as shown in Fig. 1, therefore, two bands are expected for the momentum Cut 2. As we described in the manuscript, “This band is not observed in the measured data in Fig. 1i because it is weak presumably due to photoemission matrix element effect.”

7. The Bogoliubov back-bending band should be visible even above Fermi level. But in this paper we could not observe this back-bending band.

ARPES measures only the occupied state. To see a limited energy region above the Fermi level, ARPES measurements should be done at a high temperature to induce enough thermal excitations. It is usually difficult to observe the Bogoliubov back-bending band above the Fermi level because of two contradicting requirements: Low temperature is required to have strong Bogoliubov back-bending bands while high temperature is required to have enough thermal excitation to see states above the Fermi level. As a compromise, only in a limited temperature window that is below T_c but at a relatively high temperature one may observe the Bogoliubov back-bending band above the Fermi level. We can observe the Bogoliubov back-bending band above the Fermi level in our single-layer FeSe/STO films, as shown in the two new subfigures added in Fig. S1. Fig. S1e shows the band structure measured along the momentum Cut 1 at 41 K in which the Bogoliubov back-bending band above the Fermi level can be observed. Fig. S1f shows the EDC at k_F in Fig. S1e which is consistent with the particle-hole symmetry.

To address the comment from the Referee, we have added two new subfigures in the revised version in Fig. S1 to show the Bogoliubov back-bending band observed above the

Fermi level.

There is a problem with data analysis regarding the superconducting gap.

*1. The data show that the pseudogap is open above T_c . However, 84 K is an overstatement.
2. I understood well that symmetrization of the spectrum is not a good idea in this paper. However, I don't understand why authors need to use the symmetrized spectrum to obtain $T_c=64K$. In fact, the temperature at which the gap is closed differs between the inner band and the outer band, if we use symmetrized spectra. (Fig. S4)*

As we explained before, our combined results, including the direct inspection of the back-bending band in Fig. 3a, its quantitative analysis in Fig. S8b and Fig. 3d, direct inspection on the development of the peak-dip structure in Fig. 2e and Fig. 3c and new analysis of gap opening in Fig. 2f, all indicate that superconductivity pairing occurs at ~ 83 K.

We agree with the Referee that “symmetrization of the spectrum is not a good idea in this paper”. The reason why we used the symmetrized spectrum to obtain $T_c=64K$ was just to reinforce the point. The gap closing temperatures obtained from the inner and outer bands are different if we use the symmetrization method. This is unreasonable and it indicates that the symmetrization method is not reliable.

In order to avoid possible misunderstanding, we follow the Referee’s advice to remove the gap from the EDC symmetrization procedure in Fig. 2f, and make some corresponding change in the main text.

3. I understood well that the most accurate way is to divide the superconducting gap by FD, rather than the symmetrized spectra. However, it (Fig. 2b) closes the superconducting gap at around 58 K. This is the temperature that is obvious to everyone. This is the temperature at which everyone can clearly say that the superconducting gap is closed around 58K.

The Referee has stated clearly that “symmetrization of the spectrum is not a good idea in this paper”. So the Referee should no longer use the symmetrization of the spectrum in Fig. 2b to obtain gap opening temperature because it is not reliable.

4. It is an arbitrary data arrangement to divide the spectrum divided by FD by the spectrum of 103K. This is not a good idea because there is another band that rises to the right nearby (Fig.2c), and if you divide by this, it naturally looks like there is a pseudo-gap, as in Fig.2d, e. I think this is an artificial gap.

We do not agree with the Referee’s claim that “*I think this is an artificial gap*”. It is a common practice in spectroscopic measurements to divide the spectra at low temperature by a spectrum in normal state to remove the effect of background in order to highlight the temperature-induced changes. First, the EDCs show systematic change with temperature.

All the spectra at low temperatures are divided by the same spectrum at 103 K. The peak-dip structure starts to develop at 83 K for both Cut 1 (Fig. 2d,e) and Cut 2 (Fig. 3b,c) measurements, and it gets stronger with decreasing temperature. Second, the dip structure is obvious and nearly symmetrical with respect to the Fermi level. The PDD (peak-dip-difference) shows a systematic decrease when the temperature warms up to 83 K. Third, the EDCs at 93 K do not show the dip structure (Fig. 2d,e and Fig. 3b,c). If “it naturally looks like there is a pseudo-gap” as the Referee claimed, it is difficult to understand why there is no dip structure in the 93 K data. We have carried out a simple simulation by dividing one EDC in normal state by another EDC in normal state, it is impossible to create a dip structure that is nearly symmetrical about the Fermi level. Therefore, 83 K is a definite turning point. All these results indicate that the peak-dip structure we observed in EDCs is caused by intrinsic superconductivity pairing, but not “an artificial gap” as the Referee claimed.

Reviewer #2 (Remarks to the Author):

In the manuscript entitled “Spectroscopic Evidence of Superconductivity Pairing at 83 K in Single-Layer FeSe/SrTiO₃ Films” by Xu et al., the authors present high-resolution ARPES studies of single layer FeSe/STO that provide the first ARPES evidence of a superconductivity-related gap at temperatures as high as 83 K in this system. Moreover, they observe the clear formation of Bogoliubov quasiparticle bands up to surprisingly high binding energies of near 100 meV. Lastly, they report the temperature dependence of the observed gap which they find to deviate greatly from the standard BCS behavior in the region between 64-83 K. The authors suggest that this region could either correspond to a pseudogap due to superconducting fluctuating Cooper pairs or to a true ordered state, therefore providing ARPES proof of superconductivity with $T_c=83K$ in FeSe/STO.

The mechanism and the associated phenomenology of high- T_c superconductivity in single layer FeSe/STO is arguably at the forefront of current research in the field. Despite the intensive research, reports of T_c 's surpassing the 60-65 K limit are sparse and indeed ARPES observations confirming superconductivity at even higher temperatures have been highly anticipated since the reported superconductivity up to 100 K in this system a few years back that was based on resistivity data.

The work by Xu et al. is therefore timely, groundbreaking and has potential to attract broad interest. Their ARPES results capturing so clearly the formation of Bogoliubov bands over such a wide range of energies and temperatures are very convincing. The evidence of gap formation at temperatures above 64 K seems solid. I am not against proposing that a pseudogap may be the reason for the anomalous temperature dependence reported for “region 2” but I am not convinced either that this is more probable than actual long-range pairing in this region. In this respect, the authors could consider improving the discussion section. Regardless of the cuprate analogy, the reported results are interesting on their own right and I believe they will inspire further

theoretical and experimental studies for this system.

There are a few points that need to be addressed. These concern mostly clarifying aspects of the procedure followed by the authors regarding the non-symmetrized ARPES spectra, providing more details for the BCS fits and making contact with available theory predictions that are directly relevant to the presented data.

- It appears that the gap magnitude (for the same k -point) is systematically found to be larger when using the “normalized EDC/FD” method instead of the symmetrized EDC method, even for low temperatures (e.g. see Fig.2f).

How do the authors justify this difference? For example, it has been shown that the estimated gap from ARPES should not depend on whether the symmetrization procedure of the ARPES spectra is followed or not, since the overall shape of the spectra may be asymmetric but the coherence peaks remain particle-hole symmetric (see [PRB 98, 094509 (2018)]).

Is the same effect observed for points on Cut 2? It would be helpful if the authors included a plot similar to that of Fig.2f for a point on the Cut 2 path.

In Figs 2g,3e and 4c,d the uncertainty in the the points located at the 64-83 K region (region 2) seems significant. The “Peak Dip Difference” in Fig.2g for example may very well be zero above 64 K. Similarly, the “Peak Height” and “Dip Height” may be converging to a single value at around 5 for high temperatures (as the blue and black lines seem to do in Fig.2g) but these two curves could also be bifurcating to two different values at high temperature, given the quite large allowed error for the data.

While it does seem that there exists a gap feature, the $T > 64$ K data of Fig.2e look “too noisy” and quite difficult to reliably extract from them the above mentioned quantities such as the “Peak Height” etc.

Perhaps additional plots including more data from Fermi surface points and/or more details on the post processing of the data could help further strengthen the authors case.

- In the supplementary, the authors include their results for Bi2212 in order to provide a validity check to their “normalized EDC/FD” method. Their Bi2212 data are indeed in good agreement with previous findings.

However and in contrast to Fig.2f, Fig.S5g shows that the gap converges to the same value at low temperatures. In fact, both methods shown in Fig.S5g give very similar data even for temperatures up to T_c , again in contrast to what is observed in FeSe/STO. Regardless of these differences, the three quantities shown in Fig.S5h seem to follow a T -dependence trend that is very similar to that observed for FeSe/STO (Figs.2g, 3e).

How do the authors explain the similarity between Figs.2g,3e and Fig.S5h but the difference between Fig.2f and Fig.S5g?

- The authors say: “For this momentum Cut 2, in the normal state, one electron band is observed above the Fermi level (black line in Fig. 1l) while the other band is expected from the established electronic structure that is above the Fermi level with its bottom

barely touching the Fermi level (dashed blue line in Fig. 11).”

The electronic energy band structure of single layer FeSe/STO has proven particularly difficult to calculate with standard bandstructure methods. Do the authors have any particular reference in mind? Especially since according to their ARPES data, these bands are not found by experiment. What is the established electronic structure for single-layer FeSe/STO that the authors mention?

- Connected with the above, the authors mention that they calculated the superconducting band structure using the BCS formula $E_k = \sqrt{\xi_k^2 + \Delta_k^2}$. I understand that they set Δ_k as an isotropic gap with chosen value so as to fit the experiment. What is the form of ξ_k ? Is this calculated and how? How do the authors' fits and experimental data compare with previously calculated ARPES data (see e.g. [New J. Phys. 18 022001 (2016)], [PRB 97, 060501 (2018)])?

- Superconducting pairing features in FeSe/STO up to energies close to 100meV were predicted theoretically in [PRB 97, 060501 (2018)]. I suggest that the authors compare their findings with previous such predictions regarding ARPES spectra of FeSe/STO.

- It needs to be pointed out that the pseudogap in the case of the cuprates has been studied exhaustively and its existence established by many different probes. A characteristic feature of the pseudogap in the ARPES results of cuprates is the observed Fermi arc where only part of the Fermi surface is gapped out above T_c . Another well known feature is the almost temperature independent gap above T_c seen by tunneling spectroscopy.

Are the authors aware of or expect such similar signatures also in the case of FeSe/STO?

- Is there any reasoning as to why region 2 cannot correspond to actual long range pairing, given that previous experiments ([Nature Mater. 14, 285-289 (2015)]) provide support for T_c 's up to 100 K?

Response to Reviewer #2

We thank Reviewer #2 for the careful reviewing of our paper and his/her constructive comments and suggestions to improve our paper.

1. The work by Xu et al. is therefore timely, groundbreaking and has potential to attract broad interest. Their ARPES results capturing so clearly the formation of Bogoliubov bands over such a wide range of energies and temperatures are very convincing. The evidence of gap formation at temperatures above 64 K seems solid. I am not against proposing that a pseudogap may be the reason for the anomalous temperature dependence reported for “region 2” but I am not convinced either that this is more probable than actual long-range pairing in this region. In this respect, the authors could consider improving the discussion section. Regardless of the cuprate analogy, the reported results are interesting on their own right and I believe they will inspire further

theoretical and experimental studies for this system.

We agree with the Referee that there is little convincing evidence to show that pseudogap is more probable than actual long-range pairing in region 2.

To follow Referee's suggestion, we modified the discussion section by changing "We propose that the former may correspond to superconductivity fluctuation while the latter may correspond to the phase coherence of Cooper pairs." into "There are two possibilities to assign the two regions. In the first case, the former 64-83 K region may correspond to superconductivity fluctuation while the latter region below 64 K may correspond to the phase coherence of Cooper pairs. In the second case, actual long-range pairing is realized in both regions. Further work is needed to distinguish between these two scenarios."

2. It appears that the gap magnitude (for the same k-point) is systematically found to be larger when using the "normalized EDC/FD" method instead of the symmetrized EDC method, even for low temperatures (e.g. see Fig.2f). How do the authors justify this difference? For example, it has been shown that the estimated gap from ARPES should not depend on whether the symmetrization procedure of the ARPES spectra is followed or not, since the overall shape of the spectra may be asymmetric but the coherence peaks remain particle-hole symmetric (see [PRB 98, 094509 (2018)]).

The gap size difference obtained between the symmetrized EDCs and normalized EDC/FD can be seen from Fig. R3. There is an additional peak around 20 meV in the EDC/FD measured at 103 K (black line in Fig. R3). When the 20K EDC/FD is divided by the 103K EDC/FD, the normalized EDC/FD at 20 K (red line in Fig. R3) shows an energy shift of the peak position about 1 meV compared with the original 20K EDC/FD and 20 K symmetrized EDC. The effect is small. This is why the gap size obtained from normalized EDC/FD is higher than that from the symmetrized EDCs, as shown in the original Fig. 2f.

Fig. R3. Comparison of the EDCs before and after being normalized by the 103 K EDC. The green line represents the symmetrized EDC measured at 20 K at the Fermi momentum K_{F_OR} , as marked in Fig. 2. The blue line represents original 20 K EDC divided by Fermi-Dirac function. The red line represents the 20K EDC/FD divided by the 103 K EDC/FD (black line). For comparison, the 20K EDCs are normalized to have similar peak height and value at the Fermi level.

As we explained in the manuscript, the coexisting two peaks make the EDC symmetrization unreliable in getting the gap size. Instead, it is more reliable to extract the gap size from normalized EDC/FD by removing the effect from the background in the normal state. Therefore, to avoid possible misunderstanding, we have removed the gap size obtained from the symmetrized EDCs in Fig. 2f in the revised manuscript.

3. Is the same effect observed for points on Cut 2? It would be helpful if the authors included a plot similar to that of Fig. 2f for a point on the Cut 2 path.

Following the Referee's suggestion, we have added the result on Cut 2 in the revised manuscript in Fig. S7e which shows the gap size extracted from the normalized EDC/FD in Fig. S7d. The result is consistent with that obtained for Cut 1 in Fig. 2f.

4. In Figs 2g, 3e and 4c, d the uncertainty in the points located at the 64-83 K region (region 2) seems significant. The "Peak Dip Difference" in Fig. 2g for example may very well be zero above 64 K. Similarly, the "Peak Height" and "Dip Height" may be converging to a single value at around 5 for high temperatures (as the blue and black lines seem to do in Fig. 2g) but these two curves could also be bifurcating to two different values at high temperature, given the quite large allowed error for the data.

While it does seem that there exists a gap feature, the $T > 64$ K data of Fig. 2e look "too noisy" and quite difficult to reliably extract from them the above mentioned quantities such as the "Peak Height" etc.

Perhaps additional plots including more data from Fermi surface points and/or more details on the post processing of the data could help further strengthen the authors case.

The sample quality we measured has been greatly improved compared with all the previous measurements, signified by the clear observation of band-splitting near M point, and first clear observation of the Bogoliubov back-bending band extending up to 100 meV. We also tried to use the best possible resolution to take high quality ARPES data. The weak signal between 64~83K is most likely intrinsic. The signal is intrinsically weak above T_c even in Bi2212 which has been proven to have a pseudogap, as shown in Fig. S6h in revised version.

Our conclusion on the 83K superconductivity pairing comes from the direct observation of the Bogoliubov back-bending band in Fig. 3. In addition, whether there is or is not a peak-dip structure, with particle-hole symmetry, developed at the Fermi level is more fundamental than the peak height or dip height to determine the superconductivity pairing

temperature. The normalized EDCs for Cut 1 (Fig. 2d,e) and Cut 2 (Fig. 3b,c) show systematic change with temperature. It is clear that the peak-dip structure starts to develop at 83 K in both measurements, and it get stronger with decreasing temperature. The extracted peak height and dip height, although they have relatively large uncertainty, are consistent with the results obtained from the direct observation of Bogoliubov back-bending band, and the development and temperature evolution of the peak-dip structure near the Fermi level.

5. In the supplementary, the authors include their results for Bi2212 in order to provide a validity check to their “normalized EDC/FD” method. Their Bi2212 data are indeed in good agreement with previous findings. However and in contrast to Fig. 2f, Fig. S5g shows that the gap converges to the same value at low temperatures. In fact, both methods shown in Fig.S5g give very similar data even for temperatures up to T_c , again in contrast to what is observed in FeSe/STO. Regardless of these differences, the three quantities shown in Fig.S5h seem to follow a T -dependence trend that is very similar to that observed for FeSe/STO (Figs.2g, 3e). How do the authors explain the similarity between Figs.2g,3e and Fig.S5h but the difference between Fig.2f and Fig.S5g?

As we explained before, the gap size difference obtained between the symmetrized EDCs and normalized EDC/FD in single-layer FeSe/STO films can be seen from Fig. R3, and replotted in Fig. R4a for comparison. In FeSe/STO films, because of double-peak structure, there is an additional peak at around 20 meV in the EDC/FD measured at 103 K (black line in Fig. R3 and R4a). When the 20K EDC/FD is divided by the 103K EDC/FD, the normalized EDC/FD at 20 K (red line in Fig. R3) shows an energy shift of the peak position about 1 meV compared with the original 20K EDC/FD and 20 K symmetrized EDC. The effect is small. This is why the gap size obtained from normalized EDC/FD is higher than that from the symmetrized EDCs, as shown in the original Fig. 2f.

On the other hand, in Bi2212, there is only one band in the measured momentum area. The EDC/FD measured at 180 K (black line in Fig. R4b) has only one peak around the Fermi level. There is little effect on the peak position and peak shape when the 15K EDC/FD is divided by the 180K EDC/FD. That is why, in this case, the gap sizes obtained from the symmetrized EDCs and from the normalized EDC/FD are very similar.

Fig. R4. Comparison of single-layer FeSe/STO (a) and Bi2212 (b) on the EDCs before and after being normalized by the normal state EDC. In both panels, the green line represents the symmetrized EDCs. The blue line represents original EDC divided by Fermi-Dirac function. The red line represents the EDC/FD divided by the normal state EDC/FD (black line).

6. The authors say: “For this momentum Cut 2, in the normal state, one electron band is observed above the Fermi level (black line in Fig. 1l) while the other band is expected from the established electronic structure that is above the Fermi level with its bottom barely touching the Fermi level (dashed blue line in Fig. 1l).”

The electronic energy band structure of single layer FeSe/STO has proven particularly difficult to calculate with standard band structure methods. Do the authors have any particular reference in mind? Especially since according to their ARPES data, these bands are not found by experiment. What is the established electronic structure for single-layer FeSe/STO that the authors mention?

Here “the established electronic structure” refers to the band structure of the single-layer FeSe/STO we obtained from our ARPES measurements, not from theoretical calculations. From the Fermi surface mapping in Fig. 1a and related band structures, we can get the electronic structure of the occupied state below the Fermi level for the single-layer FeSe/STO films which consists of two equivalent ellipse-like electron pockets that are perpendicular to each other. From the Bogoliubov back-bending bands we observed at low temperature in the superconducting state, we can extract the electronic structure for the unoccupied state in a limited energy range above the Fermi level, as illustrated in Fig. S2 and Fig. S3. Combining these results we obtained in the occupied and unoccupied states, we can construct a quantitative three-dimensional electronic structure near the M point, as shown in Fig. R5.

Fig. R5. Three-dimensional electronic structure near the M point constructed from our ARPES measurements.

7. Connected with the above, the authors mention that they calculated the superconducting band structure using the BCS formula $E_k = \sqrt{\xi_k^2 + \Delta_k^2}$. I understand that they set Δ_k as an isotropic gap with chosen value so as to fit the experiment. What is the form of ξ_k ? Is this calculated and how? How do the authors' fits and experimental data compare with previously calculated ARPES data (see e.g. [New J. Phys. 18 022001 (2016)], [PRB 97, 060501 (2018)])?

- Superconducting pairing features in FeSe/STO up to energies close to 100 meV were predicted theoretically in [PRB 97, 060501 (2018)]. I suggest that the authors compare their findings with previous such predictions regarding ARPES spectra of FeSe/STO.

In our paper, ξ_k is obtained directly from our ARPES measurements at a high temperature, as shown in Fig. S1. Fig. S1b shows the band structure measured at 83 K along the momentum Cut 1. Fig. S1c shows the corresponding second derivative image from Fig. S1b. The band splitting can be observed in Fig. S1c. These data are used to extract the energy band, ξ_k , at 83 K.

Following the Referee's suggestion, we have added the detailed form of curves shown in Fig. S2d in Supplementary Materials. We also add discussions in the revised manuscript "which is very similar to the calculated band structure above E_F [30]", and "which is consistent with the theoretical predictions of the superconducting pairing features in single layer FeSe/ SrTiO₃ films up to energies close to 100 meV [30]".

8. It needs to be pointed out that the pseudogap in the case of the cuprates has been studied exhaustively and its existence established by many different probes. A characteristic feature of the pseudogap in the ARPES results of cuprates is the observed Fermi arc where only part of the Fermi surface is gapped out above T_c . Another well known feature is the almost temperature independent gap above T_c seen by tunneling spectroscopy.

Are the authors aware of or expect such similar signatures also in the case of FeSe/STO?

As far as we know, there is no ARPES observation of the Fermi arc or tunneling measurement of nearly temperature-independent gap above T_c reported in FeSe/STO. Our present work provides a good opportunity to further study along this direction.

9. Is there any reasoning as to why region 2 cannot correspond to actual long range pairing, given that previous experiments ([Nature Mater. 14, 285-289 (2015)]) provide support for T_c 's up to 100 K?

In our paper, we provide spectroscopic evidence of superconductivity pairing at 83K in single-layer FeSe/SrTiO₃ films. We further divided the temperature region into 64-83K Region 2 and below 64K Region 1 according to their behaviors. We do not have conclusive evidence to decide whether Region 2 corresponds to actual long range pairing or the formation of a pseudogap. We think further experiments combining *in situ* resistivity and ARPES measurements on the same high-quality FeSe/STO samples are needed.

Summary of changes:

1. Following the comment from Reviewer #1, we add two new subfigures in Fig. S1 to show the Bogoliubov back-bending band observed above the Fermi level.
2. Following the comment from Reviewer #1, we remove the gap size from the EDC symmetrization procedure (black line in Fig. 2f) and made some corresponding modifications in the main text on page 6, line 164: “To extract the energy gap, we follow the usual procedure to get symmetrized EDCs (Fig. 2b) [32], and obtain the gap size (Δ) from the peak position. From the symmetrized EDCs, the dip structure at the Fermi level represents a gap opening; it disappears around 64 K for the Fermi momentum of the outer Fermi surface, k_{F_OR} .”
3. To clarify on the nature of the back-bending band, as the Reviewer #1 mentioned, we have added detailed analysis of the back-bending band in new Fig. S4, and related discussions about the back-bending band in the Supplementary Materials.
4. Following the comment from Reviewer #2, on page 12, line 341, we modified the discussion section by changing “We propose that the former may correspond to superconductivity fluctuation while the latter may correspond to the phase coherence of Cooper pairs.” into “There are two possibilities to assign the two regions. In the first case, the former 64-83 K region may correspond to superconductivity fluctuation while the latter region below 64 K may correspond to the phase coherence of Cooper pairs. In the second case, actual long-range pairing is realized in both regions. Further work is needed to distinguish between these two scenarios.”
5. Following Reviewer #2’s suggestion, we have added the result on Cut 2 in the revised manuscript in Fig. S7e which shows the gap size extracted from the normalized EDC/FD in Fig. S7d.
6. Following Reviewer #2’s suggestion, we have added the detailed form of curves shown in Fig. S2d in Supplementary Materials. We also add discussions in the revised manuscript “which is very similar to the calculated band structure above E_F [30]” (page 5, line120), and “which is consistent with the theoretical predictions of the superconducting pairing features in single layer FeSe/ SrTiO₃ films up to energies close to 100 meV [30]” (page 8, line 226).

REVIEWER COMMENTS

Reviewer #1 (Remarks to the Author):

I am satisfied with the new analysis in the revised manuscript. I recommend to publish in Nature Communications.

Reviewer #2 (Remarks to the Author):

The authors have addressed several of my comments satisfyingly. I have no doubt that the resolution and the sample quality in this work is high. The quality of their data for the Bogoliubov band is unprecedented.

However, as the authors discuss, applying the same technique as in cuprates to identify a possible high temperature gap may be troublesome due to the interference of the second band in FeSe/SrTiO₃ that is absent in Bi2212. My main concern at this point is whether the authors tackle this problem in a way that leads to reliable interpretation of part of their results. Specifically, how their choice of one of the two peaks at -15meV influences the conclusions about the high temperature gap value.

There remain thus a couple of issues for which the authors need to provide convincing arguments. Below I focus only on these points, I accept the rest of the authors' responses not mentioned here.

2. My previous comment:

It appears that the gap magnitude (for the same k-point) is systematically found to be larger when using the "normalized EDC/FD" method instead of the symmetrized EDC method, even for low temperatures (e.g. see Fig.2f). How do the authors justify this difference? For example, it has been shown that the estimated gap from ARPES should not depend on whether the symmetrization procedure of the ARPES spectra is followed or not, since the overall shape of the spectra may be asymmetric but the coherence peaks remain particle-hole symmetric (see [PRB 98, 094509 (2018)]).

Authors' response:

The gap size difference obtained between the symmetrized EDCs and normalized EDC/FD can be seen from Fig. R3. There is an additional peak around 20 meV in the EDC/FD measured at 103 K (black line in Fig. R3). When the 20K EDC/FD is divided by the 103K EDC/FD, the normalized EDC/FD at 20 K (red line in Fig. R3) shows an energy shift of the peak position about 1 meV compared with the original 20K EDC/FD and 20 K symmetrized EDC. The effect is small. This is why the gap size obtained from normalized EDC/FD is higher than that from the symmetrized EDCs, as shown in the original Fig. 2f.

Fig. R3. Comparison of the EDCs before and after being normalized by the 103 K EDC. The green line represents the symmetrized EDC measured at 20 K at the Fermi momentum K_F _OR, as marked in Fig. 2. The blue line represents original 20 K EDC divided by Fermi-Dirac function. The red line represents the 20K EDC/FD divided by the 103 K EDC/FD (black line). For comparison, the 20K EDCs are normalized to have similar peak height and value at the Fermi level.

As we explained in the manuscript, the coexisting two peaks make the EDC symmetrization unreliable in getting the gap size. Instead, it is more reliable to extract the gap size from normalized EDC/FD by removing the effect from the background in the normal state. Therefore, to avoid possible misunderstanding, we have removed the gap size obtained from the symmetrized EDCs in Fig. 2f in the revised manuscript.

My comment:

I am having trouble following some of the authors' reasoning. How can the peak at 20meV above the Fermi level of the 103K EDC affect the double-peak structure near 15meV below the Fermi level when dividing the two spectra?

Apart from the above, I do see that due to the smooth and monotonic shape of the 103K curve near -15meV, the highest of the two peaks of the original 20K EDC/FD curve is suppressed more after dividing, so that in the normalized 20K EDC (red line) the highest of the two peaks appears at higher binding energy (more to the left).

As far as I understand, the authors take the peak with the larger height to measure the peak-dip-difference (PDD) that gives them the gap. Therefore, since normalizing the 20K EDC changes the relative peak height, the gap after this procedure appears larger.

The authors ought to provide convincing arguments as to why they choose the highest peak in their analysis.

Given the available information, I cannot agree with this authors' choice. The highest peak does not necessarily give the location of the superconducting coherence peak in a multiband system (e.g. see fig.4d of [PRB 98, 094509 (2018)]). It is more reasonable to attribute the superconducting coherence peak to the peak closer to the Fermi level; thus, choose this peak for the authors' analysis, instead. This choice should also cure the peculiar disagreement between the gap obtained from the original 20K EDC (blue) and the normalized one (red) at low temperature.

For low temperature, the 1meV shift in the gap magnitude may seem small but such shift may become important at high temperatures where the gap is much smaller.

How do the results change (e.g. Fig.2g) when the peak to the right (lowest peak around -15meV) is used in the analysis, instead? This point needs to be clarified.

I do not find the removal of the symmetrized data from Fig.2f a good idea at this point.

3. My previous comment:

Is the same effect observed for points on Cut 2? It would be helpful if the authors included a plot similar to that of Fig.2f for a point on the Cut 2 path.

Authors' response:

Following the Referee's suggestion, we have added the result on Cut 2 in the revised manuscript in Fig. S7e which shows the gap size extracted from the normalized EDC/FD in Fig. S7d. The result is consistent with that obtained for Cut 1 in Fig. 2f.

My comment:

The inclusion of Cut 2 is a welcome addition, however, what I wanted to see was the comparison between symmetrized and normalized data. Subsequently, a comparison with similar results for Cut 1.

6. My previous comment:

The authors say: "For this momentum Cut 2, in the normal state, one electron band is observed above the Fermi level (black line in Fig. 1I) while the other band is expected from the established electronic structure that is above the Fermi level with its bottom barely touching the Fermi level (dashed blue line in Fig. 1I)."

The electronic energy band structure of single layer FeSe/STO has proven particularly difficult to calculate with standard band structure methods. Do the authors have any particular reference in mind? Especially since according to their ARPES data, these bands are not found by experiment. What is the established electronic structure for single-layer FeSe/STO that the authors mention?

Authors' response:

Here “the established electronic structure” refers to the band structure of the single-layer FeSe/STO we obtained from our ARPES measurements, not from theoretical calculations. From the Fermi surface mapping in Fig. 1a and related band structures, we can get the electronic structure of the occupied state below the Fermi level for the single-layer FeSe/STO films which consists of two equivalent ellipse-like electron pockets that are perpendicular to each other. From the Bogoliubov back-bending bands we observed at low temperature in the superconducting state, we can extract the electronic structure for the unoccupied state in a limited energy range above the Fermi level, as illustrated in Fig. S2 and Fig. S3. Combining these results we obtained in the occupied and unoccupied states, we can construct a quantitative three-dimensional electronic structure near the M point, as shown in Fig. R5.

Fig. R5. Three-dimensional electronic structure near the M point constructed from our ARPES measurements.

My comment:

The authors have clarified this point. Yet, I think that it would be more accurate if the authors say “the bandstructure established in this work” instead of just “established” which causes confusion. Also, I think it is misleading to refer to the “calculated” bandstructure in the manuscript. The word “extracted” is more suitable.

Reviewer #3 (Remarks to the Author):

The manuscript of Xu et al. without any doubts presents highest quality ARPES data on single-layer FeSe films. From their initial report in Nature Materials eight years ago, the group remains a leading photoemission team in this field. The reviewers have done an excellent job of scrutinizing many important details of the work, both in terms of its scientific relevance to the field and technical issues, and the authors have done a good job of responding. Therefore, I will not add anything to this.

I think the most unusual and, at first glance, confusing feature in the data is not even the strength of the Bogoliubov backbending dispersion. Rather, it is its unusual intensity pattern. A textbook example dictates that the spectral weight should decrease away from the (k_F, E_F) point. In contrast, the features in Figs. 1d and 1j appear to gain spectral weight near $(-0.9 \pi/a, -80 \text{ meV})$ and $(-1.8 \pi/a, -80 \text{ meV})$, respectively. Such features are strongly similar to replicas due to some kind of reconstruction rather than superconductivity.

This apparent controversy can be resolved by remembering that normally no one looks at the details of the normal state behavior of dispersion ABOVE the Fermi level, but assumes it to be linear over a wide energy range, as in the textbook example. The band structure of iron-based superconductors is more complicated and it has been shown (arXiv:1106.4584) that merging branches of “bogoliubons” can indeed lead to an enhancement of the spectral weight at higher energies. The authors can confirm that also for the single-layer FeSe film, the dispersion forming the outer electron pocket along the diagonal merges very quickly with the electron-like band in the center of the BZ (Z point in FeSe) and thus may be responsible for the observed paradox, since it will be “reflected” by pairing to the occupied part.

Otherwise, I fully support the publication.

Response to Reviewer's Comments

Reviewer #1 (Remarks to the Author):

I am satisfied with the new analysis in the revised manuscript. I recommend to publish in Nature Communications.

Response to Reviewer #1

We thank Reviewer #1 for reviewing our paper again and recommending its publication in Nature Communications.

Reviewer #2 (Remarks to the Author):

The authors have addressed several of my comments satisfyingly. I have no doubt that the resolution and the sample quality in this work is high. The quality of their data for the Bogoliubov band is unprecedented.

However, as the authors discuss, applying the same technique as in cuprates to identify a possible high temperature gap may be troublesome due to the interference of the second band in FeSe/SrTiO₃ that is absent in Bi2212. My main concern at this point is whether the authors tackle this problem in a way that leads to reliable interpretation of part of their results. Specifically, how their choice of one of the two peaks at -15meV influences the conclusions about the high temperature gap value.

There remain thus a couple of issues for which the authors need to provide convincing arguments. Below I focus only on these points, I accept the rest of the authors' responses not mentioned here.

2. My previous comment:

It appears that the gap magnitude (for the same k-point) is systematically found to be larger when using the "normalized EDC/FD" method instead of the symmetrized EDC method, even for low temperatures (e.g. see Fig.2f). How do the authors justify this difference? For example, it has been shown that the estimated gap from ARPES should not depend on whether the symmetrization procedure of the ARPES spectra is followed or not, since the overall shape of the spectra may be asymmetric but the coherence peaks remain particle-hole symmetric (see [PRB 98, 094509 (2018)]).

Authors' response:

The gap size difference obtained between the symmetrized EDCs and normalized EDC/FD can be seen from Fig. R3. There is an additional peak around 20 meV in the EDC/FD measured at 103 K (black line in Fig. R3). When the 20K EDC/FD is divided by the 103K

EDC/FD, the normalized EDC/FD at 20 K (red line in Fig. R3) shows an energy shift of the peak position about 1 meV compared with the original 20K EDC/FD and 20 K symmetrized EDC. The effect is small. This is why the gap size obtained from normalized EDC/FD is higher than that from the symmetrized EDCs, as shown in the original Fig. 2f.

Fig. R3. Comparison of the EDCs before and after being normalized by the 103 K EDC. The green line represents the symmetrized EDC measured at 20 K at the Fermi momentum K_{F_OR} , as marked in Fig. 2. The blue line represents original 20 K EDC divided by Fermi-Dirac function. The red line represents the 20K EDC/FD divided by the 103 K EDC/FD (black line). For comparison, the 20K EDCs are normalized to have similar peak height and value at the Fermi level.

As we explained in the manuscript, the coexisting two peaks make the EDC symmetrization unreliable in getting the gap size. Instead, it is more reliable to extract the gap size from normalized EDC/FD by removing the effect from the background in the normal state. Therefore, to avoid possible misunderstanding, we have removed the gap size obtained from the symmetrized EDCs in Fig. 2f in the revised manuscript.

My comment:

I am having trouble following some of the authors' reasoning. How can the peak at 20meV above the Fermi level of the 103K EDC affect the double-peak structure near 15meV below the Fermi level when dividing the two spectra?

Apart from the above, I do see that due to the smooth and monotonic shape of the 103K curve near -15meV, the highest of the two peaks of the original 20K EDC/FD curve is suppressed more after dividing, so that in the normalized 20K EDC (red line) the highest of the two peaks appears at higher binding energy (more to the left).

As far as I understand, the authors take the peak with the larger height to measure the peak-dip-difference (PDD) that gives them the gap. Therefore, since normalizing the 20K EDC changes the relative peak height, the gap after this procedure appears larger.

The authors ought to provide convincing arguments as to why they choose the highest peak in their analysis.

Given the available information, I cannot agree with this authors' choice. The highest peak does not necessarily give the location of the superconducting coherence peak in a multiband system (e.g. see fig.4d of [PRB 98, 094509 (2018)]). It is more reasonable to attribute the superconducting coherence peak to the peak closer to the Fermi level; thus, choose this peak for the authors' analysis, instead. This choice should also cure the peculiar disagreement between the gap obtained from the original 20K EDC (blue) and the normalized one (red) at low temperature.

For low temperature, the 1meV shift in the gap magnitude may seem small but such shift may become important at high temperatures where the gap is much smaller.

How do the results change (e.g. Fig.2g) when the peak to the right (lowest peak around -15meV) is used in the analysis, instead? This point needs to be clarified.

I do not find the removal of the symmetrized data from Fig.2f a good idea at this point.

3. My previous comment:

Is the same effect observed for points on Cut 2? It would be helpful if the authors included a plot similar to that of Fig.2f for a point on the Cut 2 path.

Authors' response:

Following the Referee's suggestion, we have added the result on Cut 2 in the revised manuscript in Fig. S7e which shows the gap size extracted from the normalized EDC/FD in Fig. S7d. The result is consistent with that obtained for Cut 1 in Fig. 2f.

My comment:

The inclusion of Cut 2 is a welcome addition, however, what I wanted to see was the comparison between symmetrized and normalized data. Subsequently, a comparison with similar results for Cut 1.

6. My previous comment:

The authors say: "For this momentum Cut 2, in the normal state, one electron band is observed above the Fermi level (black line in Fig. 1l) while the other band is expected from the established electronic structure that is above the Fermi level with its bottom barely touching the Fermi level (dashed blue line in Fig. 1l)."

The electronic energy band structure of single layer FeSe/STO has proven particularly difficult to calculate with standard band structure methods. Do the authors have any particular reference in mind? Especially since according to their ARPES data, these bands are not found by experiment. What is the established electronic structure for single-layer FeSe/STO that the authors mention?

Authors' response:

Here "the established electronic structure" refers to the band structure of the single-layer FeSe/STO we obtained from our ARPES measurements, not from theoretical calculations. From the Fermi surface mapping in Fig. 1a and related band structures, we can get the electronic structure of the occupied state below the Fermi level for the single-layer FeSe/STO films which consists of two equivalent ellipse-like electron pockets that are perpendicular to each other. From the Bogoliubov back-bending bands we observed at low temperature in the superconducting state, we can extract the electronic structure for the unoccupied state in a limited energy range above the Fermi level, as illustrated in Fig. S2 and Fig. S3. Combining these results we obtained in the occupied and unoccupied states, we can construct a quantitative three-dimensional electronic structure near the M point, as shown in Fig. R5.

Fig. R5. Three-dimensional electronic structure near the M point constructed from our ARPES measurements.

My comment:

The authors have clarified this point. Yet, I think that it would be more accurate if the authors say “the band structure established in this work” instead of just “established” which causes confusion.

Also, I think it is misleading to refer to the “calculated” band structure in the manuscript. The word “extracted” is more suitable.

Response to Reviewer #2

We thank Reviewer #2 for the careful reviewing of our paper again and his/her constructive comments and suggestions to improve our paper.

1. I am having trouble following some of the authors’ reasoning. How can the peak at 20meV above the Fermi level of the 103K EDC affect the double-peak structure near 15meV below the Fermi level when dividing the two spectra?

Apart from the above, I do see that due to the smooth and monotonic shape of the 103K curve near -15meV, the highest of the two peaks of the original 20K EDC/FD curve is suppressed more after dividing, so that in the normalized 20K EDC (red line) the highest of the two peaks appears at higher binding energy (more to the left).

As far as I understand, the authors take the peak with the larger height to measure the peak-dip-difference (PDD) that gives them the gap. Therefore, since normalizing the 20K EDC changes the relative peak height, the gap after this procedure appears larger.

The authors ought to provide convincing arguments as to why they choose the highest peak in their analysis.

Given the available information, I cannot agree with this authors’ choice. The highest peak does not necessarily give the location of the superconducting coherence peak in a multiband system (e.g. see fig.4d of [PRB 98, 094509 (2018)]). It is more reasonable to attribute the superconducting coherence peak to the peak closer to the Fermi level; thus, choose this peak for the authors’ analysis, instead. This choice should also cure the peculiar disagreement between the gap obtained from the original 20K EDC (blue) and the normalized one (red) at low temperature.

For low temperature, the 1meV shift in the gap magnitude may seem small but such shift may become important at high temperatures where the gap is much smaller.

How do the results change (e.g. Fig.2g) when the peak to the right (lowest peak around -15meV) is used in the analysis, instead? This point needs to be clarified.

I do not find the removal of the symmetrized data from Fig.2f a good idea at this point.

The Reviewer’s difficulty to follow our reasoning comes from a misunderstanding. The EDC measured at 20 K at the Fermi momentum K_F_OR in Fig. 2 consists of a SINGLE peak, not two peaks as the Reviewer considered. This is because K_F_OR is on the right side of the two bands, as seen in the inset of Fig. 2a.

For convenience, we show the related figure (Fig. R3 in our previous Response) again in Fig. R1 below. Here the EDC below the Fermi level consists of only one single peak although there is some data fluctuation near the peak top region. Also we took the overall peak position, not the position of the highest point, to determine the gap size. For example, for the normalized EDC (red line in Fig. R1 below), the position of the highest point is 16 meV but the gap value we took in the paper is the peak position, 15 meV.

As suggested by the Reviewer, we add the gap size extracted from the symmetrized data back to Fig. 2f in the revised version.

Fig. R1. Comparison of the EDCs before and after being normalized by the 103 K EDC. The green line represents the symmetrized EDC measured at 20 K at the Fermi momentum K_{F_OR} , as marked in Fig. 2b. The blue line represents original 20 K EDC divided by Fermi-Dirac function. The red line represents the 20K EDC/FD divided by the 103 K EDC/FD (black line). For comparison, the 20K EDCs are normalized to have similar peak height and value at the Fermi level.

3. My previous comment:

Is the same effect observed for points on Cut 2? It would be helpful if the authors included a plot similar to that of Fig.2f for a point on the Cut 2 path.

Authors' response:

Following the Referee's suggestion, we have added the result on Cut 2 in the revised manuscript in Fig. S7e which shows the gap size extracted from the normalized EDC/FD in Fig. S7d. The result is consistent with that obtained for Cut 1 in Fig. 2f.

My comment:

The inclusion of Cut 2 is a welcome addition, however, what I wanted to see was the comparison between symmetrized and normalized data. Subsequently, a comparison with similar results for Cut 1.

Following the Reviewer's suggestion, we add symmetrized EDCs for Cut 2 in Fig. S7c and the gap size extracted from the symmetrized EDCs in Fig. S7f in the revised version.

6. My previous comment:

The authors say: "For this momentum Cut 2, in the normal state, one electron band is observed above the Fermi level (black line in Fig. 11) while the other band is expected from the established electronic structure that is above the Fermi level with its bottom barely touching the Fermi level (dashed blue line in Fig. 11)."

The electronic energy band structure of single layer FeSe/STO has proven particularly difficult to calculate with standard band structure methods. Do the authors have any particular reference in mind? Especially since according to their ARPES data, these bands are not found by experiment. What is the established electronic structure for single-layer FeSe/STO that the authors mention?

Authors' response:

Here "the established electronic structure" refers to the band structure of the single-layer FeSe/STO we obtained from our ARPES measurements, not from theoretical calculations. From the Fermi surface mapping in Fig. 1a and related band structures, we can get the electronic structure of the occupied state below the Fermi level for the single-layer FeSe/STO films which consists of two equivalent ellipse-like electron pockets that are perpendicular to each other. From the Bogoliubov back-bending bands we observed at low temperature in the superconducting state, we can extract the electronic structure for the unoccupied state in a limited energy range above the Fermi level, as illustrated in Fig. S2 and Fig. S3. Combining these results we obtained in the occupied and unoccupied states, we can construct a quantitative three-dimensional electronic structure near the M point, as shown in Fig. R5.

Fig. R5. Three-dimensional electronic structure near the M point constructed from our ARPES measurements.

My comment:

The authors have clarified this point. Yet, I think that it would be more accurate if the authors say "the bandstructure established in this work" instead of just "established" which causes confusion.

Also, I think it is misleading to refer to the "calculated" bandstructure in the manuscript. The word "extracted" is more suitable.

Following the Reviewer's suggestion, we change the expression of "the established

electronic structure” to “the band structure established in this work”. And we also change “calculated” to “extracted” in the revised version.

Reviewer #3 (Remarks to the Author):

The manuscript of Xu et al. without any doubts presents highest quality ARPES data on single-layer FeSe films. From their initial report in Nature Materials eight years ago, the group remains a leading photoemission team in this field. The reviewers have done an excellent job of scrutinizing many important details of the work, both in terms of its scientific relevance to the field and technical issues, and the authors have done a good job of responding. Therefore, I will not add anything to this.

I think the most unusual and, at first glance, confusing feature in the data is not even the strength of the Bogoliubov backbending dispersion. Rather, it is its unusual intensity pattern. A textbook example dictates that the spectral weight should decrease away from the (k_F , E_F) point. In contrast, the features in Figs. 1d and 1j appear to gain spectral weight near $(-0.9\pi/a, -80\text{ meV})$ and $(-1.8\pi/a, -80\text{ meV})$, respectively. Such features are strongly similar to replicas due to some kind of reconstruction rather than superconductivity.

This apparent controversy can be resolved by remembering that normally no one looks at the details of the normal state behavior of dispersion ABOVE the Fermi level, but assumes it to be linear over a wide energy range, as in the textbook example. The band structure of iron-based superconductors is more complicated and it has been shown (arXiv:1106.4584) that merging branches of "bogoliubons" can indeed lead to an enhancement of the spectral weight at higher energies. The authors can confirm that also for the single-layer FeSe film, the dispersion forming the outer electron pocket along the diagonal merges very quickly with the electron-like band in the center of the BZ (Z point in FeSe) and thus may be responsible for the observed paradox, since it will be "reflected" by pairing to the occupied part.

Otherwise, I fully support the publication.

Response to Reviewer #3

We thank Reviewer #3 for the careful reviewing of our paper and his/her constructive comments and suggestions to improve our paper. We thank the Reviewer for capturing the high quality of our data and the significance of our work, and recommending its publication in Nature Communications.

The Reviewer observed that “*the features in Figs. 1d and 1j appear to gain spectral weight near $(-0.9 \pi/a, -80 \text{ meV})$ and $(-1.8 \pi/a, -80 \text{ meV})$ ”*. But as we showed in the previous Response and the added Fig. S4, it is not true that the back-bending band gains weight near $(-0.9 \pi/a, -80 \text{ meV})$ and $(-1.8 \pi/a, -80 \text{ meV})$. For convenience, we show related figure (Fig. S4 in Supplementary) as Fig. R2 below. As seen in Fig. R2d, at the binding energy of 80 meV, the back-bending band looks stronger than those at lower binding energies (60 meV and 40 meV) because it is sitting on a stronger background from the central broad peak. As shown in Fig. R2e, after subtracting the background from the central broad peak, the net spectral weight of the back-bending band at the binding energy of 80 meV is weaker than those at lower binding energies. As shown in Fig. R2b, quantitative analysis of the spectral weight along the back-bending band is consistent with the BCS theory. Therefore, it is not necessary to invoke new scenario to explain the back-bending band we observed.

Fig. R2. Analysis of the Bogoliubov back-bending band. (a) Bogoliubov back bending band formed at 20 K for the momentum Cut 2. The black dashed line represents the initial band in the normal state while the blue solid line represents the back-bending band. (b) Coherence factor deduced from the normal state band and the back-bending band at 20 K in (a). The intensity of the back-bending band from EDC analysis (c) and MDC analysis (e) is plotted. The intensity for point A at the Fermi momentum is taken as 0.5 to normalize the EDC intensity for the point B and C. The intensity of point C is taken as the same from both the EDC and MDC analysis to normalize the spectral intensity of the D and E points. (c) EDCs along cuts #1, #2 and #3 as shown in (a). The spectral weight of the back-bending band is marked. (d) Momentum distribution curves (MDCs) along cuts #4, #5 and #6, corresponding to binding energies of 40 meV, 60 meV and 80 meV, respectively, as shown in (a). (e) Same MDCs as in (d) but are plotted with an offset for clarity. The spectral weight of the back-bending band is marked as shadow area.

Summary of changes:

1. Following the suggestion from Reviewer #2, we add the gap size extracted from the symmetrized data back to Fig. 2f in revised version.
2. Following Reviewer #2's suggestion, we add symmetrized EDCs for Cut 2 in Fig. S7c and the gap size extracted from the symmetrized EDCs in Fig. S7f in revised version.
3. Following the Reviewer's suggestion, we change the expression of "the established electronic structure" to "the band structure established in this work". And we also change "calculated" to "extracted" in the manuscript in revised version.

REVIEWERS' COMMENTS

Reviewer #2 (Remarks to the Author):

The authors have responded to all my remaining comments in a satisfying manner. I recommend their manuscript for publication in Nature Communications.

Response to Reviewer's Comments

Reviewer #2 (Remarks to the Author):

The authors have responded to all my remaining comments in a satisfying manner. I recommend their manuscript for publication in Nature Communications.

Response to Reviewer #2

We thank Reviewer #2 for reviewing our paper again and recommending its publication in Nature Communications.

.